

# Quantifying biostabilisation effects of biofilm-secreted and synthetic extracellular polymeric substances (EPS) on sandy substrate

**Wietse I. van de Lageweg\*, Stuart J. McLelland and Daniel R. Parsons**

Geography and Geology, School of Environmental Sciences, University of Hull, Cottingham Road, Kingston-Upon-Hull, HU6 7RX, United Kingdom.

\*Now at: Faculty of Geosciences, Utrecht University, The Netherlands

*Correspondence to*: Wietse I. van de Lageweg (wietse.vandelageweg@gmail.com)

**Abstract.** Microbial assemblages ('biofilms') preferentially develop at water-sediment interfaces and are
known to have a considerable influence on sediment stability and erodibility. There is potential for significant impacts on sediment transport and morphodynamics and, hence, on the longer-term evolution of coastal and fluvial environments. However, the biostabilisation effects remain poorly understood and quantified due to the inherent complexity of biofilms and the large spatial and temporal (i.e. seasonality) variations involved. Here, we use controlled laboratory tests to systematically quantify the effects of
natural biofilm colonisation as well as synthetic extracellular polymeric substances (EPS) on sediment stability. Synthetic EPS may be useful to simulate biofilm mediated biostabilisation, and potentially provide a method of speeding up time scales of physical modelling experiments investigating biostabilisation effects. We find a mean biostabilisation due to natural biofilm colonisation and development of almost four times that of the uncolonised sand. The presented cumulative probability
distribution of measured critical erosion thresholds reflects the large spatial and temporal variations generally seen in natural biostabilised environments. For identical sand, engineered sediment stability

from the addition of synthetic EPS compares well across the measured range and behaves in a linear and predictable fashion. Yet, the effectiveness of synthetic EPS to stabilise sediment is sensitive to the preparation procedure, time after application and environmental conditions such as salinity, pH and temperature. These findings are expected to improve bio-physical experimental models in fluvial and

coastal environments and provide much-needed quantification of biostabilisation to improve predictions of sediment dynamics in aquatic ecosystems.

*Keywords: Biofilm, biostabilisation, EPS, physical modelling, ecology, sediment transport*

## 1 Introduction

Micro-organisms are a fundamental feature of aquatic environments providing a range of ecosystem services (Gerbersdorf et al. 2011; Gerbersdorf and Wieprecht 2015). A large variety of microbial assemblages ('biofilms') such as micro-phytobenthos, microbial mats, flocs, aggregates and biofouling in pipes (Flemming and Wingender 2010) are representations of microbial communities in aqueous environments. The microbes in biofilms live in a self-formed matrix of glue-like and hydrated

extracellular polymeric substances (EPS) such as polysaccharides (often 40-95%), proteins (up to 60%) and minor amounts of acids, lipids and biopolymers (Decho 1990; Flemming 2011; Gerbersdorf et al. 2011). The ecosystem functions of EPS in sediment particle aggregation, increasing sediment stability, altering chemical properties to enable contaminant release or adsorption, and providing a food source for invertebrates are well established for marine environments (Decho 1990; Passow 2002; Bhaskar and

Bhosle 2006; Paterson et al. 2008), but remain less well understood for freshwater systems (Gerbersdorf et al. 2011). The ability of biofilms to stabilize sediment and protect sedimentary surfaces against erosion

is often referred to as 'biostabilisation' (cf. Paterson 1989). Biostabilisation may result from coverage by microbial mats which protects underlying sediments from fluid forces (Noffke and Paterson 2007) or from micro- to macroscopically thin biofilms that coat, bridge or permeate single grains and pore spaces with their EPS (Gerbersdorf and Wieprecht 2015) which increases sediment cohesion and increases the

entrainment threshold.

Many studies have attempted to quantify biostabilisation in a variety of environments (Paterson 1989; Dade et al. 1990; Amos et al. 1998; Tolhurst et al. 1999; Tolhurst et al. 2003; Friend et al. 2003; Friend, Collins, and Holligan 2003; Droppo et al. 2007; Righetti and Lucarelli 2007; Vignaga, Haynes, and Sloan

2012; Graba et al. 2013; Thom et al. 2015). These studies generally show a positive correlation between EPS content and sediment stability measured using an erosion threshold, although variations in space and time (Friend, Collins, and Holligan 2003; Thom et al. 2015) and between cohesive and non-cohesive sandy environments are large. Biostabilisation of coarse sand and gravel may increase the erosion threshold up to almost three times compared to abiotic sediment (Vignaga, Haynes, and Sloan 2012) while

a tenfold increase in erosion threshold compared to abiotic sediment has been reported for fine sands and cohesive sediments (Paterson 1997; Dade et al. 1990). EPS is known to add biostability in two ways: 1) by physically binding both cohesive and non-cohesive sediment grains together (see Tolhurst, Gust, and Paterson (2002) for low-temperature scanning electron microscopy images of biofilm-secreted EPS strands binding sediment particles together), and 2) by molecular electrochemical interaction with

cohesive clay particles (Chenu and Guérif 1991).

Biofilm formation affects sediment erosion, transport, deposition and consolidation (Righetti and Lucarelli 2007; Gerbersdorf and Wieprecht 2015). There is, for example, evidence that diatom blooms alter estuarine sediment dynamics (Kornman and De Deckere 1998) illustrating the potential effects micro-organisms can have on system-wide sediment fluxes. At a smaller scale, the introduction of the

synthetic EPS Xanthan Gum in flume experiments investigating bedform dynamics has been shown to change bedform morphology and behaviour (Malarkey et al. 2015; Parsons et al. 2016). Changes in delta morphology and behaviour were also observed in flume experiments where EPS was added to the sediment mixture (Hoyal and Sheets 2009; Kleinhans et al. 2014). Furthermore, evidence is growing that biofilms alter their local environment by affecting hydrodynamics (Vignaga et al. 2013), since the biofilm

surface changes the bed roughness to either dampen or increase turbulence production (Gerbersdorf and Wieprecht 2015), and sometimes their protruding structures create a buffer layer between the flow and the sediment bed that can enhance settling rates (e.g. Augspurger and Küsel 2010).

The corollary of the evidence showing the impact of biofilms on sediment stability and flow behaviour is

that the inclusion of biological processes and responses is critical to modelling sediment dynamics because micro-organisms are an integral component of the functioning of water and sediment transfer systems. Predicting the potential impacts of climate change on aquatic environments and applying bio-engineering adaptation strategies like '*Building with Nature'* for coastal defence (de Vriend et al. 2015) or flood resilience (Temmerman et al. 2013) requires an understanding of i) the response of micro-

organisms to changes in climate-induced hydrodynamic forcing, and ii) the role of micro-organisms in water and sediment transfer systems. Even though it has been demonstrated that the synthetic EPS

Xanthan Gum is not a perfect analogue for natural biofilms (Perkins et al. 2004), it is useful for modelling biological interactions with sediment dynamics (e.g. Hoyal and Sheets 2009; Kleinhans et al. 2014; Malarkey et al. 2015; Parsons et al. 2016). Synthetic EPS also has the advantage that enables time scales of physical modelling experiments to be reduced and biostabilisation effects to be controlled.

The objective of this study is therefore to evaluate biostabilisation effects of existing synthetic EPS for a range of conditions commonly used in physical modelling experiments. In doing so, the study solely focusses on the sediment stabilising aspect of biofilms and does explicitly not intend to replicate and evaluate natural biofilm behaviour and effects. The development of a robust methodology and protocol

for the application and resultant impacts of synthetic EPS are expected to inform future studies seeking to introduce biological cohesion in a rapid and controlled manner. A sandy substrate was used in this study since this grain size range is most commonly used in physical models of coastal and fluvial systems to date. The specific aims of this study are to:

1. Quantify biostabilisation effects (i.e. erosion threshold) of biofilm-secreted EPS on sandy

substrates in a physical model experiment.

2. Using the same sandy substrate, quantify the biostabilisation effects of four synthetic EPS.

3. Assess the sensitivity of the biostabilisation effects of the four synthetic EPS to:

    a. The preparation procedure

    b. The time after application

c. Environmental factors that may differ between flume facilities such as salinity, pH and

       temperature

Earth **Surface**
Dynamics
Discussions

4.  Summarise the key steps and findings into a protocol informing future work on usage and expected

    biostabilisation effects.

## 2 Material and methods

This study reports on a flume experiment in which a natural biofilm is allowed to colonise a sandy

substrate. The observations made on spatial and temporal dynamics of the sediment stabilising capacity

of the natural biofilm provide a reference for auxiliary tests on the sediment stabilising capacity of

synthetic extracellular polymeric substances (EPS). The aim of the auxiliary tests was to quantify the

ability of synthetic EPS to replicate the sediment stabilising capacity of natural biofilms in a fast and

controlled manner. Below we describe the materials and methods used in both experiments.

### 2.1 Biofilm experiment

#### 2.1.1 Experimental setup and conditions

The biofilm experiment was setup in the Total Environment Simulator flume facility at the University of

Hull (Figure 1). Nine parallel channels were constructed for colonisation. Each channel was 9 m long,

0.48 m wide and contained a 0.1 m thick substrate layer. For five channels, the substrate consisted of 110

micron sand. One channel contained a coarser 1 mm sandy substrate and one channel contained a fifty-

fifty mixture of the 110 micron sand and 1 mm sand. The two remaining channels contained a patterned

substrate of alternating patches of the 110 micron sand and 1 mm sand, with different lengths of the

patches for the two channels. Here, we will focus on the five channels with the 110 micron sandy substrate

that allowed us to investigate the temporal dynamics involved in biofilm colonisation and stabilisation. Importantly, the same 110 micron sand was also used in the auxiliary tests with synthetic EPS.

Brackish water (~30 grams of salt per litre) representative of estuarine, mangrove and deltaic settings was re-circulated at a constant rate. Typical flow velocities were 0.01 – 0.05 m/s with higher flow velocities for the central channels due to the inlet conditions. Lighting consisted of ten grow lamps, positioned in two parallel lines of five light sources. Illuminance tests showed that the central channels received the highest light intensity (~3000 lux) with lower intensities towards marginal channels (~1500 lux). Such light intensities correspond to an overcast day. The grow lamps were alternately switched on and off for 12 hours, although the experiment was never completely dark because fluorescent lighting around the flume remained switched on during the night for safety purposes.

The total experimental duration was seven weeks. During the first two weeks, the biofilm community was allowed to establish and no measurements were made. In this two-week period, inoculation of the flume proceeded from using eutrophic waste water from the local aquarium and by placing rocks with a biofilm sampled from the local Humber estuary in the flume. Then, weekly measurements of EPS content and sediment entrainment were made over a five-week period. The measurements required partial draining of the flume and therefore about 20% of the water volume was replaced weekly with new waste water from the aquarium. This also ensured that high nutrient levels were maintained during the entire experimental duration. Soil samples from the top 0.01 m of every channel were taken to determine the EPS content from (see section 2.1.2 Determination of EPS content for details on methodology to determine EPS from

soil samples). In total, 80 soil samples were collected in this way. Similarly, two sediment entrainment

measurements for each channel were collected using the Cohesive Strength Meter (CSM) erosion device

(see section 2.2 Cohesive Strength Meter (CSM) erosion device for details on the CSM erosion device).

In total, 61 successful CSM measurements were made.

## 2.1.2 Determination of EPS content

EPS content was calculated using the phenol sulphuric acid method, employing colour differences to

determine the amount of carbohydrates (Dubois et al. 1956). The methodology can be subdivided into

two main steps. First, 1.5 grams of each soil sample were weighted and placed into 15 ml centrifuge tubes.

Five millilitres of 0.5Mm Ethylene Diamine Tetraacetic acid (EDTA) solution was added to each tube.

The sediment-EDTA solution was then centrifuged at 5000 rpm. Following centrifuging, the supernatants

were pooled and a placed in a 50 ml centrifuge tube. This was repeated two more times. Then, 35 ml of

ethanol was added to the 15 ml of supernatant and left overnight.

The second step started with a 30-minutes centrifuge at 5000 rpm of the ethanol-supernatant solution.

Then, the precipitate was dissolved in 1 millilitre of MilliQ water from which the amount of carbohydrates

was measured using the phenol sulphuric acid method. This method uses a set of standards to produce a

calibration curve. In this study, the standards had glucose concentrations ranging between 0 µg/ml and

40 µg/ml. Standards were produced by mixing 200 µl of the respective glucose solution with 200 µl of

phenol solution and 1 millilitre of concentrated sulphuric acid. The samples were prepared according to





the same procedure, but by replacing the glucose solution with the aqueous solution. Finally, the

absorbance was measured using a spectrophotometer at 490 nm. Using the glucose calibration curve, the

measured absorbance was converted to a carbohydrate amount that was assumed equal to the amount of

EPS. Dry weight of the soil sample was used to calculate the EPS content.

## 2.2 Cohesive Strength Meter (CSM) erosion device

The CSM is an erosion device (https://partrac-csm.com/) that allows for quantification of sediment

entrainment thresholds and erosion rates in the laboratory as well as in the field across a variety of

environments (Paterson 1989; Tolhurst et al. 1999; Tolhurst, Gust, and Paterson 2002). The CSM uses a

vertical jet of water that impinges on the sediment surface generating a normal and tangential stress at the

interface. These stresses were converted to a critical horizontal shear stress ($\tau_c$) according to the calibrated

formulation:

$$\tau_c = 66.67 \cdot \left(1 - e^{\frac{-C}{310.09}}\right) - 195.28 \cdot \left(1 - e^{\frac{-C}{1622.57}}\right)$$

(1)

Where $C$ is the CSM measured vertical threshold stress (kPa).

The CSM allows 39 different test routines making it is possible to vary the jet pulse duration, the pressure

increments and the maximum applied pressure. For all data reported in this study, test routine S7 was

used as it strikes a balance between fine pressure increments while reaching a high maximum pressure,

thus covering a large erosional range within the same setting. Another motivation for selection of routine

S7 is that it was used in Tolhurst et al. (2002) as well, allowing for a direct comparison between the data.

This routine starts at 0kPa, incrementing by 2 kPa per step up to 82 kPa with a jet being fired for 1 second.

## 2.3 Sediment sample tests with synthetic EPS

The effect of varying amounts of four different types of synthetic EPS on the sediment entrainment threshold and erosion behaviour was tested. The four different EPS: Xanthan Gum, Alginic Acid, Carrageenan and Agar were selected for their ease of availability, differences in chemical properties, and

absence of safety issues ensuring the potential for wide usage in future work. Xanthan Gum ($C_{35}H_{49}O_{29}$) is a polysaccharide commonly used as a food additive and has also been included in earlier laboratory tests (Tolhurst, Gust, and Paterson 2002; Parsons et al. 2016). Alginic Acid ($C_6H_8O_6)_n$, also known as alginate, is a carbohydrate produced by brown algae and also widely used in food. Carrageenan is a sulphate polysaccharide extracted from red seaweeds and also widely used as a food additive. We used

the Iota variety that has two sulphate groups per disaccharide ($C_{24}H_{36}O_{25}S_2$). Agar is used as a gelling agent and is obtained from the polysaccharide agarose found in some species of red algae.

A protocol similar to the one used in Tolhurst et al. (2002) was used to prepare the sediment samples for CSM testing. A control test with no EPS, and four tests with increasing EPS contents of 1.25 g, 2.5 g, 5

g and 10 g per kg of sediment were performed for the four different EPS. The required EPS amount was added to 330 ml of distilled water and mixed thoroughly by a magnetic stirrer. The EPS solution was then added to 650 g of dry 110 micron sand and mixed with an electric stirrer to distribute the EPS solution throughout the sand. The sand-EPS mixture was then poured into plastic petri dishes (5 cm diameter) to a depth of 1 cm. Irregularities on the sediment surface increase the bed roughness and stress (Tolhurst et

al. 2002), therefore care was taken to create a level surface by tapping the side of the petri dishes before

testing. All test conditions were repeated five times and all tests were performed under fully saturated

conditions.

### 2.3.1 Preparation procedure

Protocol development on the application and effects of different synthetic EPS required an assessment of

the impact of the preparation procedure on the sediment entrainment threshold. To this end, the

preparation procedure described above, referred to as 'Wet Mixing', was complemented by a preparation

procedure referred to as Dry Mixing. Both procedures used the same sand, EPS and amounts but the order

in which they were combined and mixed, was changed. In contrast to the Wet Mixing procedure, in the

Dry Mixing procedure the required amount of EPS was first added to the sand and mixed with an electric

stirrer. Then, 330 ml of distilled water was added to the dry sand-EPS mixture and a further mixing with

the electrical stirrer was performed. Note that the risk of dust formation and associated loss of EPS powder

was greater in the Dry Mixing procedure.

### 2.3.2 Environmental conditions

Protocol development on the application and effects of different EPS also required an assessment of the

impact of the different environmental conditions on the sediment entrainment threshold. As temperature,

salinity and to a lesser extent pH commonly vary between flume facilities, a sensitivity analysis on the

effectiveness of synthetic EPS to impact the sediment entrainment threshold was performed. For

temperature, tests were performed at 10° Celsius and 40 ° Celsius in addition to the control tests at room

temperature of 20° Celsius. For pH, tests were performed with a pH of 4 and a pH of 10 in addition to the

control tests of a pH of 7. For salinity, tests with a salinity of 30 ppm corresponding to brackish conditions were performed in addition to the control tests with distilled fresh water.

## 3 Results

The eutrophic water used in the experiment resulted in rapid colonisation and growth of a diatomaceous

biofilm on the substrate materials (Figure 1A). After two weeks, biofilm colonisation and growth was localised and organised into a darker stripes running parallel to the main flow (Figure 1B). Colonisation and development of the biofilm continued over the next five weeks resulting in a more widespread biofilm coverage (Figure 1C). At the end of the experiment after seven weeks, the sandy substrate in the channels was covered by a few millimetres thickness of black biofilm crust (Figure 1D). At this stage, mortality of

the biofilm had set in locally, which was illustrated by greyish patches within the black healthy biofilm that were sometimes eroded. This observation ensured that we observed the full life cycle of a diatomaceous biofilm from early colonisation to mortality and subsequent crust erosion.

Microscope investigations of the species ecology confirmed a saline environment that was dominated by

halophilous diatoms, which are common in coastal zones (Pan et al. 2004). The diverse flora was dominated by five main species: a) *Nitzschia pellucida*, b) *Nitzschia sigma*, c) *Mastogloia sp*, d) *Navicula perminuta*, and e) *Amphora pediculus*. The *Nitzschia* species are considered early colonisers (Ledger et al. 2008; Ros, Marín-Murcia, and Aboal 2009), and were indeed found primarily in the samples of the early stages of the experiment. Furthermore, all taxa were benthic rather than planktonic, as expected in

lotic conditions (Passy 2001; Schmidt et al. 2016). Some were attached, some floated around the substrate.



Also, ciliates were present and presumably eating the diatoms. Importantly, many of the species observed

were obligate and cannot tolerate freshwater, in agreement with the designed experimental conditions.

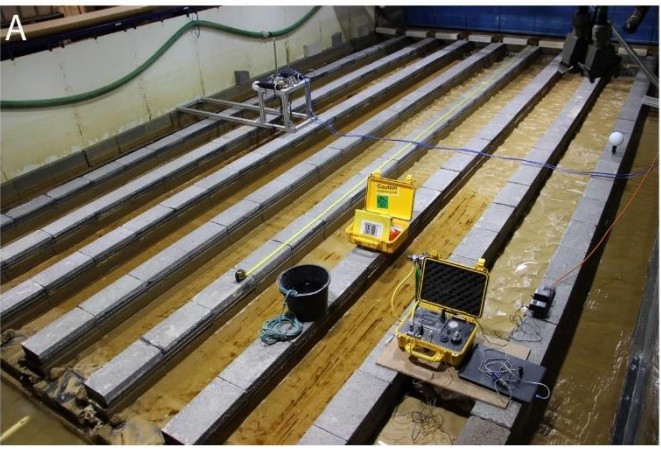
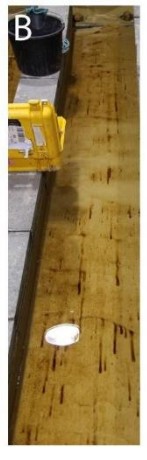
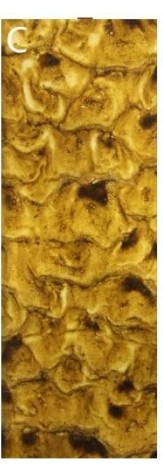
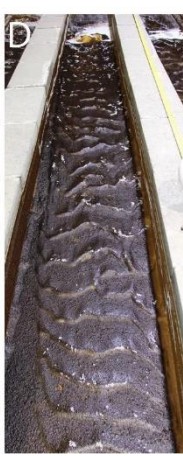

**Figure 1: Biofilm experiment in Total Environment Simulator flume facility. A) Overview of experimental setup showing nine (9) parallel channels for biofilm colonisation. Channels are 9 meters long, 0.48 m wide and contain a 0.1 m thick substrate layer consisting of uniform 110 micron sandy sediment. Also visible in the yellow cases is the CSM erosion device. Panels B) – D) show colonisation and development of a diatomaceous biofilm on the sandy substrate from early onset in (B) to a mature and dark biofilm after 6 weeks. Flow in panels A), C) and D) is towards viewer, and away from viewer in panel B).**

## 3.1 Added sediment stability from biofilm-secreted EPS

Figure 2 shows a cumulative probability distribution of the CSM sediment stability measurements made

during the flume experiment. The average shear stress entrainment threshold was 0.69 $N \cdot m^{-2}$ with a

standard deviation of 0.82 $N \cdot m^{-2}$. The distribution is highly skewed towards lower shear stresses, as

evidenced by a median shear stress entrainment threshold of 0.23 $N \cdot m^{-2}$. This median value was just

above the CSM measured entrainment threshold for the uncolonised sand of 0.18 $N \cdot m^{-2}$, which is in close

agreement with the theoretical entrainment threshold for the applied 110 micron sand of 0.15 $N \cdot m^{-2}$



(Zanke 2003). Notably, 42% of the measurements were smaller than the entrainment threshold of the uncolonised sand, even when a biofilm was clearly visible at the substrate surface. A maximum entrainment threshold of 3.84 N·m$^{-2}$ was measured, which represents a more than 21 times higher erodibility threshold compared to the uncolonised sand. Entrainment thresholds were higher in the first

5   three weeks (~ 1 N·m$^{-2}$ on average) in comparison to the last two weeks (~ 0.3 N·m$^{-2}$ on average).

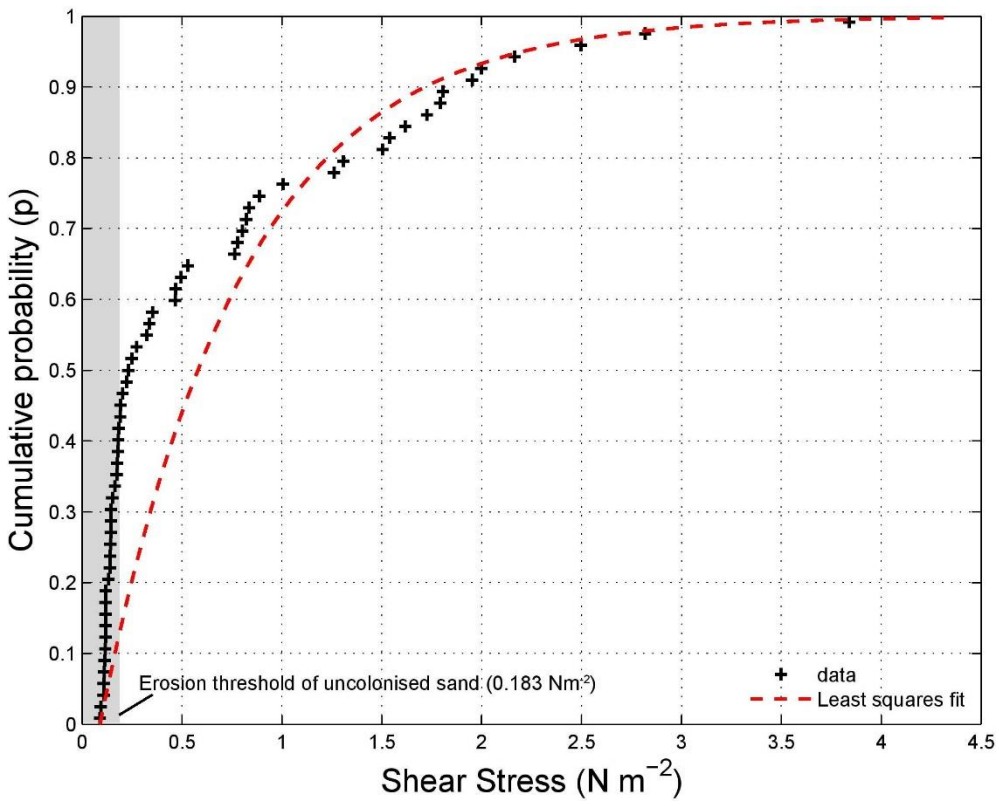

**Figure 2. Shear stress measurements made with CSM erosion device during natural biofilm growth experiment. The measurements (n = 61) are best described by a least squares exponential fit with a mean parameter $\mu$ of 0.71.**

The average carbohydrate content, here equated to EPS content, was 7.8 µg per g of sand with a

10   standard deviation of 7.8 µg per g (Figure 3). The measurements were best described by an exponential

fit with a mean parameter $\mu$ of 7.88, highlighting the skewed character of the data with many lower



content observations and fewer towards higher EPS contents. The maximum measured EPS content was

34.6 µg per g of sand. In contrast to the sediment entrainment threshold (Figure 2), the average EPS

content increased on a weekly basis from 5.6 µg per g of sand.in the first week to 11.6 µg per g of

sand.in the final week.

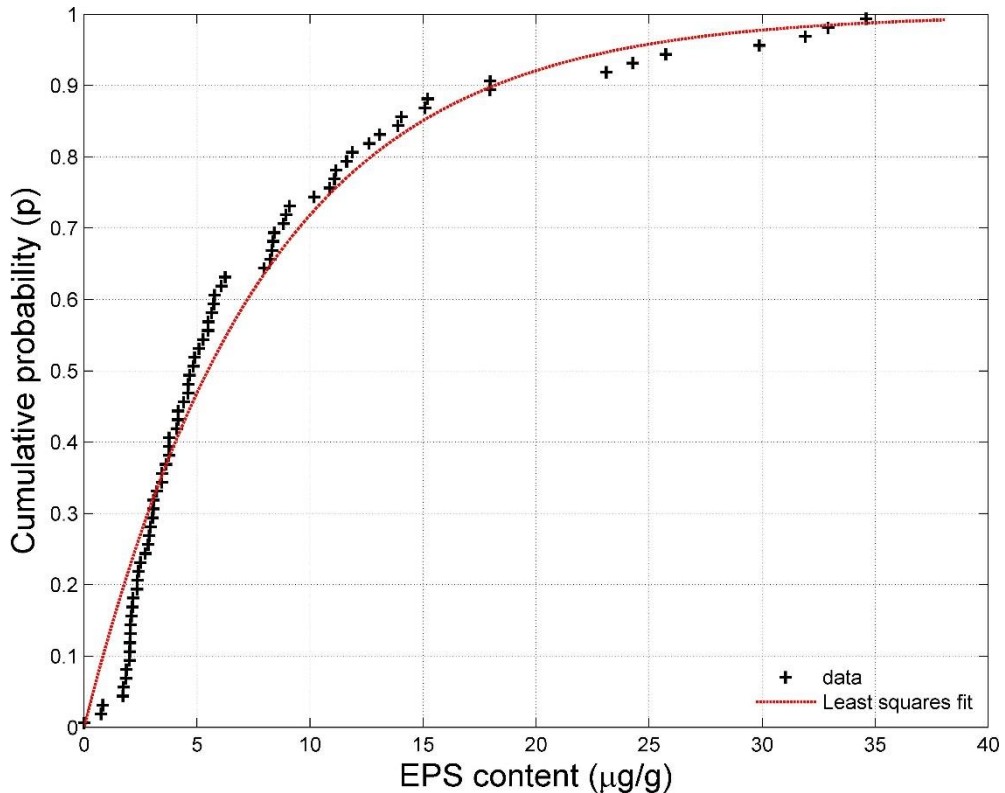

**Figure 3. Extracellular polymeric substances (EPS) content measurements made during natural biofilm growth experiment. The measurements (n = 80) are best described by a least squares exponential fit with a mean parameter $\mu$ of 7.88.**

## 3.2 Added sediment stability from synthetic EPS

10   The above section 3.1 Added sediment stability from biofilm-secreted EPS illustrated that experiments

involving natural biofilms typically take multiple weeks to capture the complete life cycle. As such flume

experiments are costly, synthetic EPS has the potential to provide an effective alternative to reproduce the sediment stabilising effects on natural biofilms in a fast and controlled manner. Below, small-scale experiments are described quantifying 1) the effect of the different concentrations of four synthetic EPS, 2) the effect of the preparation procedure, and 3) the effect of environmental factors such as temperature,

5    salinity and pH. All three tests were intended to contribute towards the development of a protocol to guide the use of synthetic EPS in experiments as a synthetic surrogate to replace natural biofilms. The applied concentrations of the synthetic EPS were based on the measured EPS contents in the natural biofilm experiment (Figure 3) and reported values in the literature (Taylor, Paterson, and Mehlert 1999; Tolhurst, Gust, and Paterson 2002).

### 3.2.1 Effects of synthetic EPS content on sediment stability

The four synthetic EPS had different effects on sediment stability (Figure 4). Alginic Acid and Agar did not increase the sediment stability above the erosion threshold of the sand without EPS, for all applied concentrations. For Xanthan Gum and Carrageenan, the erosion threshold generally increased with

15    increasing EPS content (Table 1). For these EPS, the relation between the critical shear stress for erosion and EPS content was best described using linear models (Figure 4), where the slope of the linear model for Xanthan Gum (0.28) was more than double the slope of the linear model for Carrageenan (0.11).





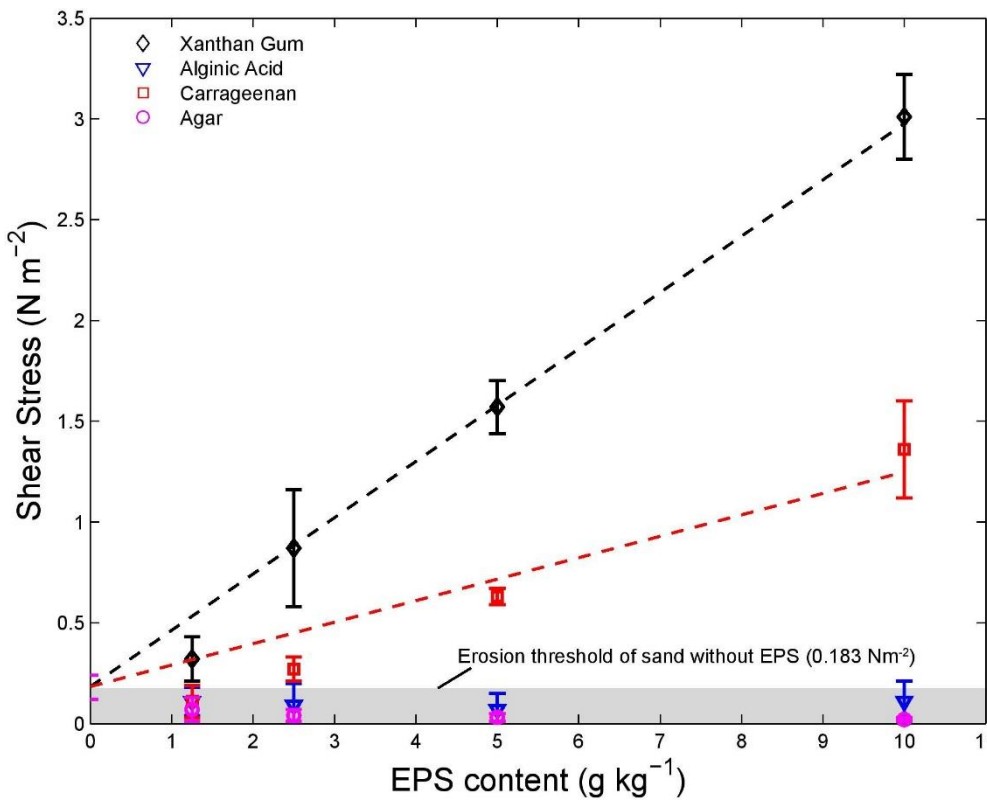

**Figure 4. The erosion thresholds of 110 micron sandy substrate with different contents for four synthetic EPS as measured with the CSM erosion device. Best fit curves were fitted using linear models for Xanthan Gum (Shear stress threshold = 0.28 EPS content + 0.18) and Carrageenan (Shear stress threshold = 0.11 EPS content + 0.18). Error bars are standard deviation from n =5 repeat measurements.**

**Table 1. Erosion thresholds for four synthetic EPS measured with the CSM erosion device.**

**Average ± St. deviation erosion threshold (N·m⁻²)**

| EPS (g·kg⁻¹) | Xanthan Gum | Carrageenan | Agar | Alginic Acid |
|---|---|---|---|---|
| **0** | 0.18 ± 0.06 | 0.18 ± 0.06 | 0.18 ± 0.06 | 0.18 ± 0.06 |
| **1.25** | 0.32 ± 0.11 | 0.11 ± 0.08 | 0.07 ± 0.06 | 0.11 ± 0.08 |



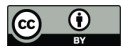

| | | | | |
|---|---|---|---|---|
| **2.5** | 0.87 ± 0.29 | 0.27 ± 0.06 | 0.04 ± 0.03 | 0.09 ± 0.11 |
| **5** | 1.57 ± 0.13 | 0.63 ± 0.04 | 0.03 ± 0.02 | 0.07 ± 0.08 |
| **10** | 3.01 ± 0.21 | 1.36 ± 0.24 | 0.02 ± 0.01 | 0.11 ± 0.10 |

### 3.2.2 Effects of preparation procedure on sediment stability

The preparation procedure adopted for adding the synthetic compounds to the sediment material had an impact on the resultant erosion threshold (Figure 5). 'Dry mixing' the synthetic EPS powder with the

5   sediment prior to adding water resulted in a higher erosion threshold than 'Wet mixing' the EPS powder with sediment in water for all tested EPS. The difference was greatest for Xanthan Gum with a 67% higher threshold for the dry mixing procedure compared to the wet mixing procedure.



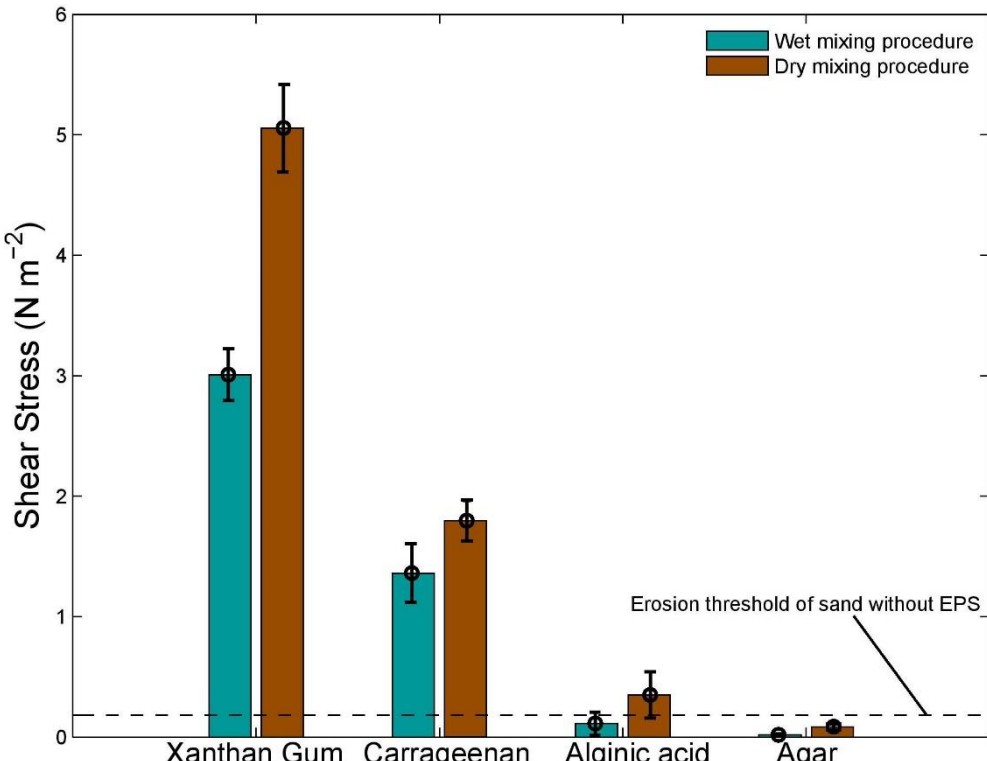

**Figure 5. The erosion thresholds as a function of the preparation procedure for four surrogates as measured with the CSM erosion device. Wet mixing involves dissolving the synthetic EPS powder in water and stir, then add sediment and mix. Dry mixing involves the addition of synthetic EPS powder to sediment and mix, then add water and stir. Error bars are standard deviation from n =5 repeat measurements.**

### 3.2.3 Temporal effects on sediment stability

Time elapsed from initial mixing also affected the sediment stabilising capacity of synthetic EPS (Figure 6). Repeat measurements after one day, seven days and fifteen days demonstrated that the erosion thresholds remained constant throughout the first week. However, the repeat measurements after fifteen days showed a decrease in the erosion threshold below the erosion threshold of sand without EPS. This



effectively meant that after about two weeks of initial application of EPS the impact on the erosion

threshold of the sediment ceased to exist.

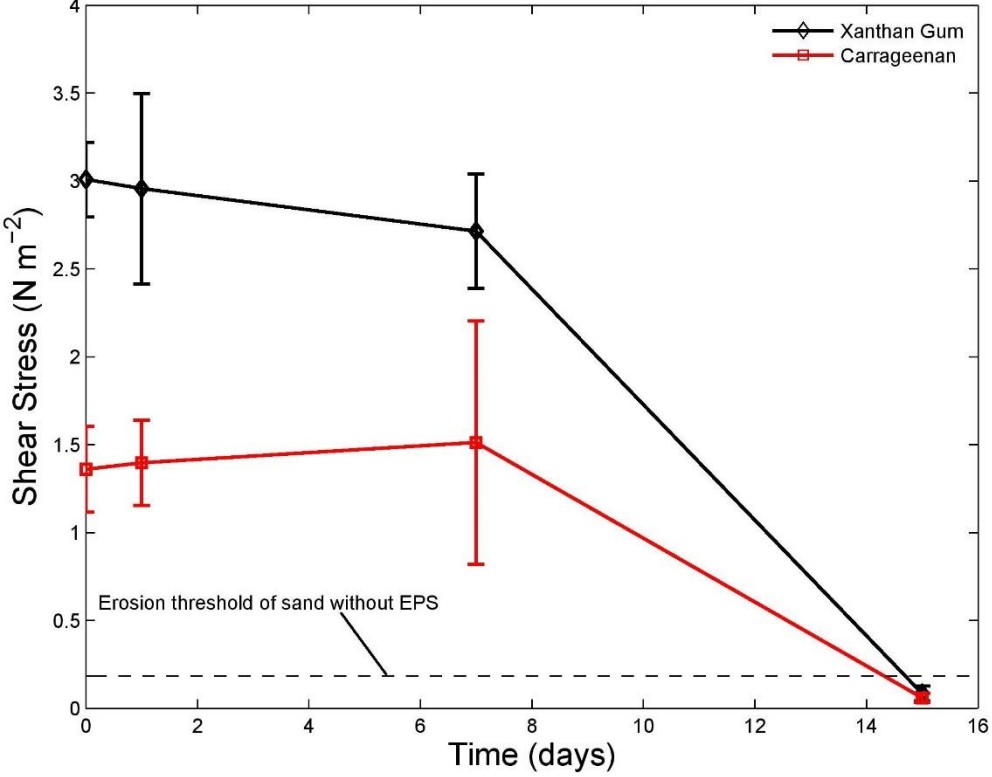

**Figure 6. The erosion thresholds as a function of time for Xanthan Gum and Carrageenan as measured with the CSM erosion device.**

5    **Error bars are standard deviation from n =3 repeat measurements.**

### 3.2.4 Effects of salinity on sediment stability

Salinity had a limited effect on the erosion thresholds (Figure 7). Saline water tended to decrease the

erosion threshold compared to freshwater conditions, though the differences are statistically insignificant

10    for all four EPS. The erosion thresholds for Alginic Acid and Agar remained below the erosion threshold

of sand without EPS independent of the salinity of the water.



This implies that the findings of this study that were mostly obtained for freshwater conditions can be extrapolated to saline conditions.

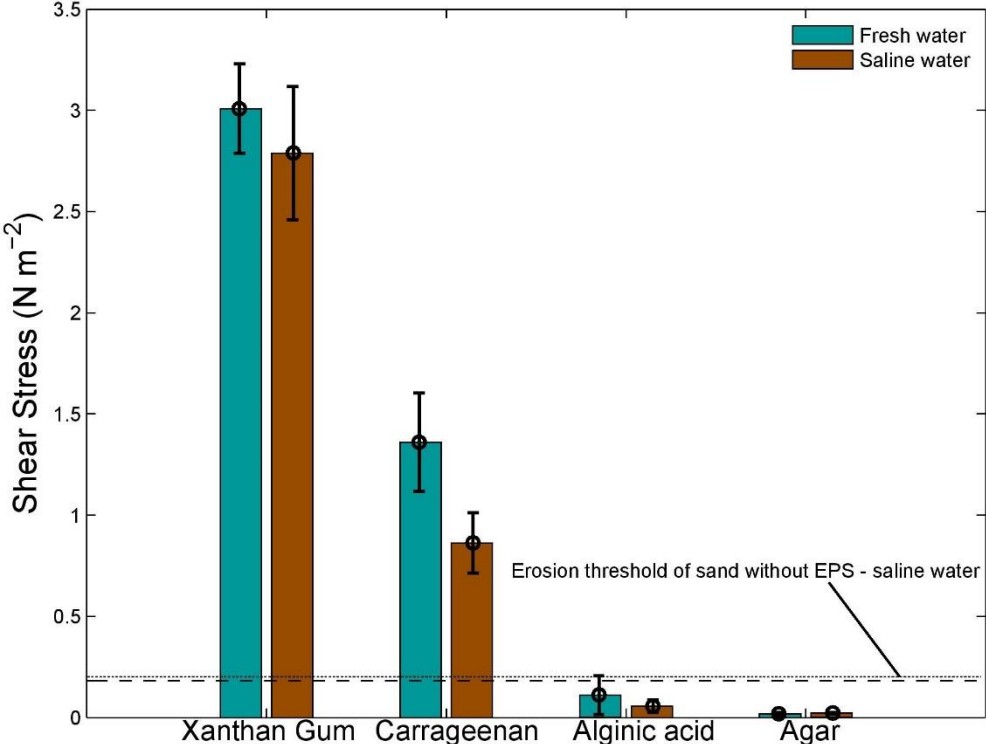

**Figure 7. The erosion thresholds as a function of salinity for four synthetic EPS as measured with the CSM erosion device. Tap water was used for the freshwater tests and a salinity of 30 ppt was used for the saline water tests. The horizontal lines correspond to the erosion thresholds of sand without EPS for freshwater (dashed) and saline water (dotted). Error bars are standard deviation from n =3 repeat measurements.**

### 3.2.5 Effects of pH on sediment stability

The pH of the applied solution had variable effects on the erosion threshold (Figure 8). An acid solution with a pH of 4 resulted in a higher erosion threshold for Xanthan Gum but in a lower threshold for Carrageenan. An alkaline solution with a pH of 10 resulted in lower erosion thresholds for Xanthan Gum



as well as Carrageenan. The erosion thresholds for Alginic Acid and Agar remained below the erosion

threshold of sand without EPS, independent of the pH of the solution.

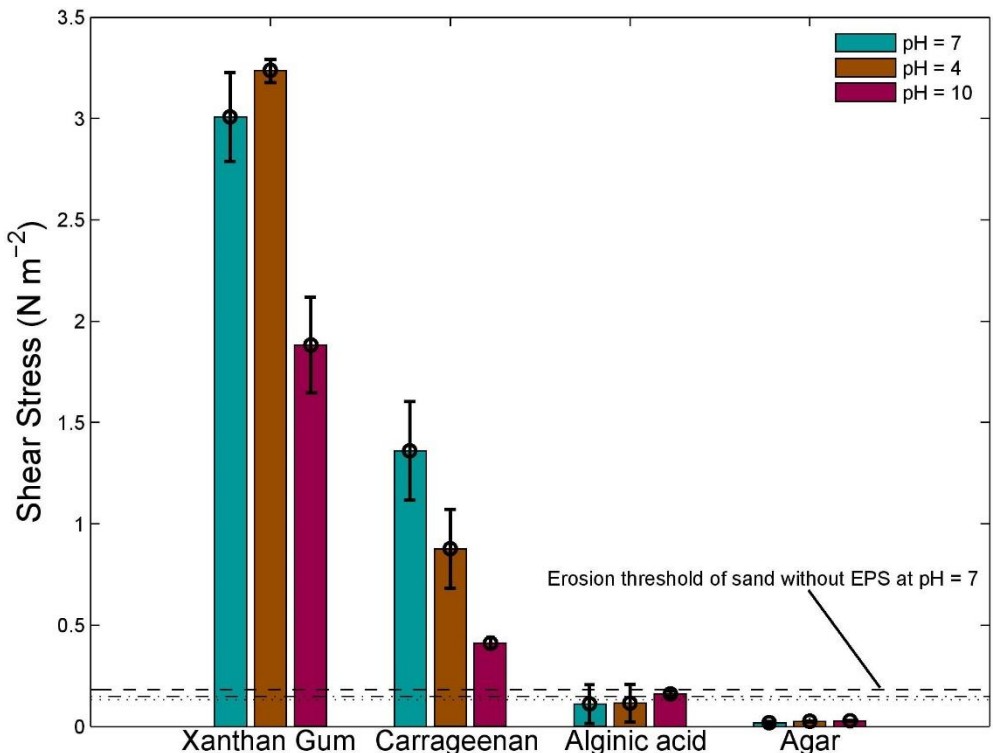

5   **Figure 8. The erosion thresholds as a function of pH for four synthetic EPS as measured with the CSM erosion device. The horizontal lines correspond to the erosion thresholds of sand without EPS for water with a pH of 7 (dashed), a pH of 4 (dotted), and a pH of 10 (dash-dotted). Error bars are standard deviation from n =3 repeat measurements.**

### 3.2.6 Effects of temperature on sediment stability

10   Temperature impacted the measured erosion thresholds (Figure 9). Both a lower temperature of 10°

Celsius and a higher temperature of 40° Celsius resulted in lower erosion thresholds. For Xanthan Gum



as well as Carrageenan, the erosion thresholds were about half under 10° Celsius and 40° Celsius test conditions compared to 20° Celsius test conditions. The erosion thresholds for Alginic Acid and Agar remained below the erosion threshold of sand without EPS independent of the temperature.

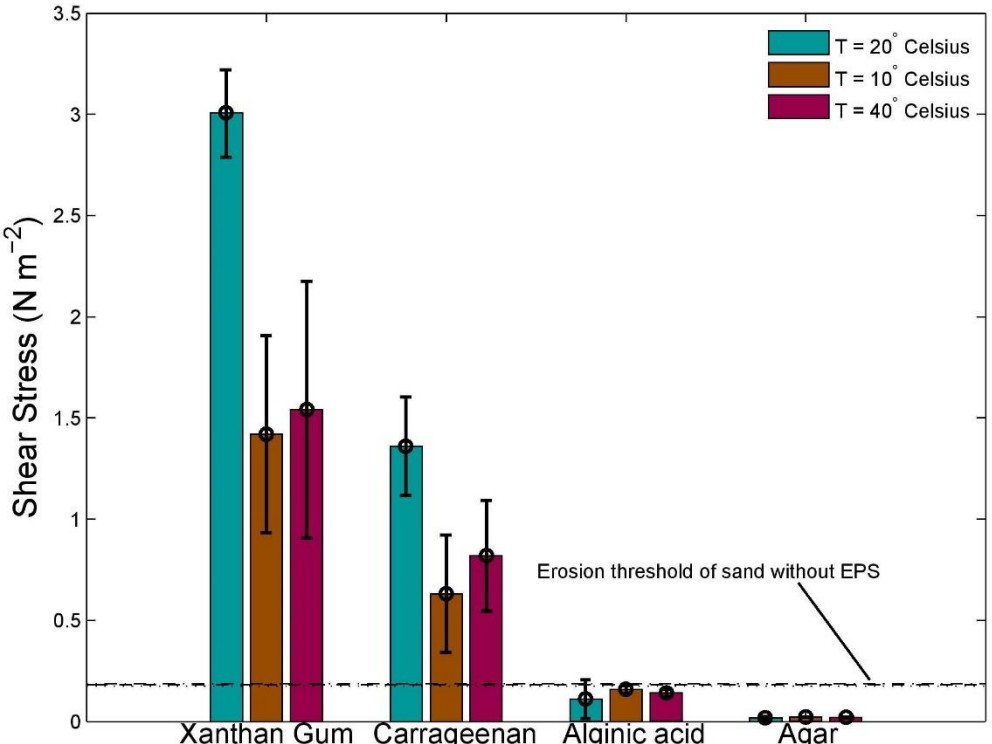

**Figure 9. The erosion thresholds as a function of temperature for four synthetic EPS as measured with the CSM erosion device. The horizontal lines correspond to the erosion thresholds of sand without EPS for a temperature of 20° Celsius (dashed), a temperature of 10° Celsius (dotted), and a temperature of 40° Celsius (dash-dotted). Error bars are standard deviation from n =3 repeat measurements.**

### 3.2.7 Synthesis of the effects of synthetic EPS on sediment stability

In summary, synthetic EPS Xanthan Gum and Carrageenan increased the erosion threshold with higher EPS content (Table 1). For these two EPS, the relation between erosion threshold and EPS content was

linear and predictable (Figure 4). In contrast, the synthetic EPS Alginic Acid and Agar did not increase

the erosion threshold (Table 1), independent of the applied concentration (Figure 4), preparation

procedure (Figure 5) or environmental condition such as salinity, pH and temperature. Yet, this study

demonstrated that the preparation procedure, environmental conditions and time impacted on the resultant

erosion threshold for the EPS Xanthan Gum and Carrageenan. A dry mixing procedure increased the

erosion threshold while saline water, alkaline solutions and non-room temperature test conditions of 20°

Celsius decreased the erosion thresholds. The tests also showed that the effects of adding Xanthan Gum

and Carrageenan on the erosion thresholds ceased to exist after about two weeks following initial

application (Figure 6). These findings indicate that the effectiveness of synthetic EPS to stabilise sediment

is sensitive to the applied concentration, the preparation procedure, time and environmental conditions.

**4 Discussion**

The CSM data show that addition of synthetic EPS Xanthan Gum and Carrageenan increases the critical

erosion threshold, even at low EPS concentrations (Figure 4 and Table 1). The observation that the erosion

threshold increased approximately linear with EPS content for Xanthan Gum is in agreement with the

findings reported in Tolhurst, Gust, and Paterson (2002). We find a similar linear increase in erosion

threshold with EPS content for Carrageenan, though the rate of increase is smaller compared to Xanthan

Gum. The approximately linear relation between EPS content and erosion threshold across the measured

range for Xanthan Gum and Carrageenan simplifies the prediction of biostabilisation effects due to

synthetic EPS. Two other synthetic EPS, Alginic Acid and Agar, were also tested and showed negligible

biostabilisation for any of the test conditions investigated.



Biostabilisation of the same sandy substrate due to natural biofilm colonisation and due to addition of synthetic EPS Xanthan Gum and Carrageenan compares well (Table 2). We find a mean biostabilisation due to natural biofilm colonisation and development of almost four times that of the uncolonised sand.

Such biostabilisation is within the reported range for fine sand (Dade et al. 1990; Vignaga et al. 2013). More specifically, 42% of the tested samples did not show biostabilisation compared to uncolonised sand while 10% of the measurements showed a tenfold biostabilisation relative to uncolonised sand (Figure 2). The presented cumulative probability distribution of critical erosion thresholds reflects the large spatial and temporal variations generally seen in natural biostabilised environments (Paterson 1989; Amos et al.

1998; Tolhurst et al. 1999; Tolhurst et al. 2003; Friend, Collins, and Holligan 2003). Biostabilisation due to synthetic EPS covers approximately the same range of erosion thresholds for the applied EPS contents. Xanthan Gum may be more suited to replicate the higher biostabilisation observations of natural biofilms due to the increased erosion thresholds for the highest applied content of 10 g·kg$^{-1}$. Carrageenan may be more appropriate to replicate the lower biostabilisation observations of natural biofilms due to the small

effect on erosion thresholds for low concentrations.

**Table 2. Relative biostabilisation resulting from natural biofilm and Xanthan Gum and Carrageenan synthetic EPS as measured in this study. Biostabilisation is defined relative to the erosion threshold of sand without EPS.**

|  | Uncolonised | Median | Mean | Maximum |
|---|---|---|---|---|
| **Biofilm** | 1 | 1.3 | 3.8 | 21.0 |

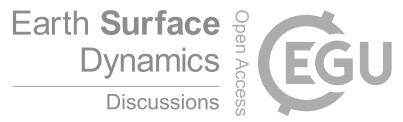

|  | 1.25 g·kg⁻¹ | 2.5 g·kg⁻¹ | 5 g·kg⁻¹ | 10 g·kg⁻¹ |
|---|---|---|---|---|
| **Xanthan Gum** | 1.7 | 4.8 | 8.6 | 16.4 |
| **Carrageenan** | 0.6 | 1.5 | 3.5 | 7.4 |

| (10 g·kg⁻¹) | Dry mix | Saline | pH = 10 | T = 10° Celsius |
|---|---|---|---|---|
| **Xanthan Gum** | 27.6 | 15.2 | 10.3 | 7.8 |
| **Carrageenan** | 9.8 | 4.7 | 2.2 | 1.6 |

The concentrations of the EPS derived from the natural biofilm experiment ($\mu g\ g^{-1}$) are about three orders of magnitude lower than the applied synthetic EPS concentrations ($mg\ g^{-1}$) to achieve the same biostabilisation effect. Two reasons may explain these differences. First, the applied phenol-sulphuric

5   acid assay measures a carbohydrate fraction of the total EPS, along with low-weight sugars that are extracted with the polymeric material (Underwood, Paterson, and Parkes 1995). Along with the sensitivity of the EPS extraction methodology to a host of conditions (Perkins et al. 2004), this may be part of the explanation for the lower EPS concentrations in the natural biofilm samples. Second, sediment sampling for EPS concentration analysis typically involved scraping off the top centimetre of the substrate.

10  However, it has been shown that EPS content in nature is highest at the sediment surface (top 200 $\mu m$) and decreases with depth (Taylor and Paterson 1998). Our sediment sampling strategy is likely to have



diluted the EPS concentration, which may offer another explanation for the lower EPS concentrations in

the natural biofilm samples.

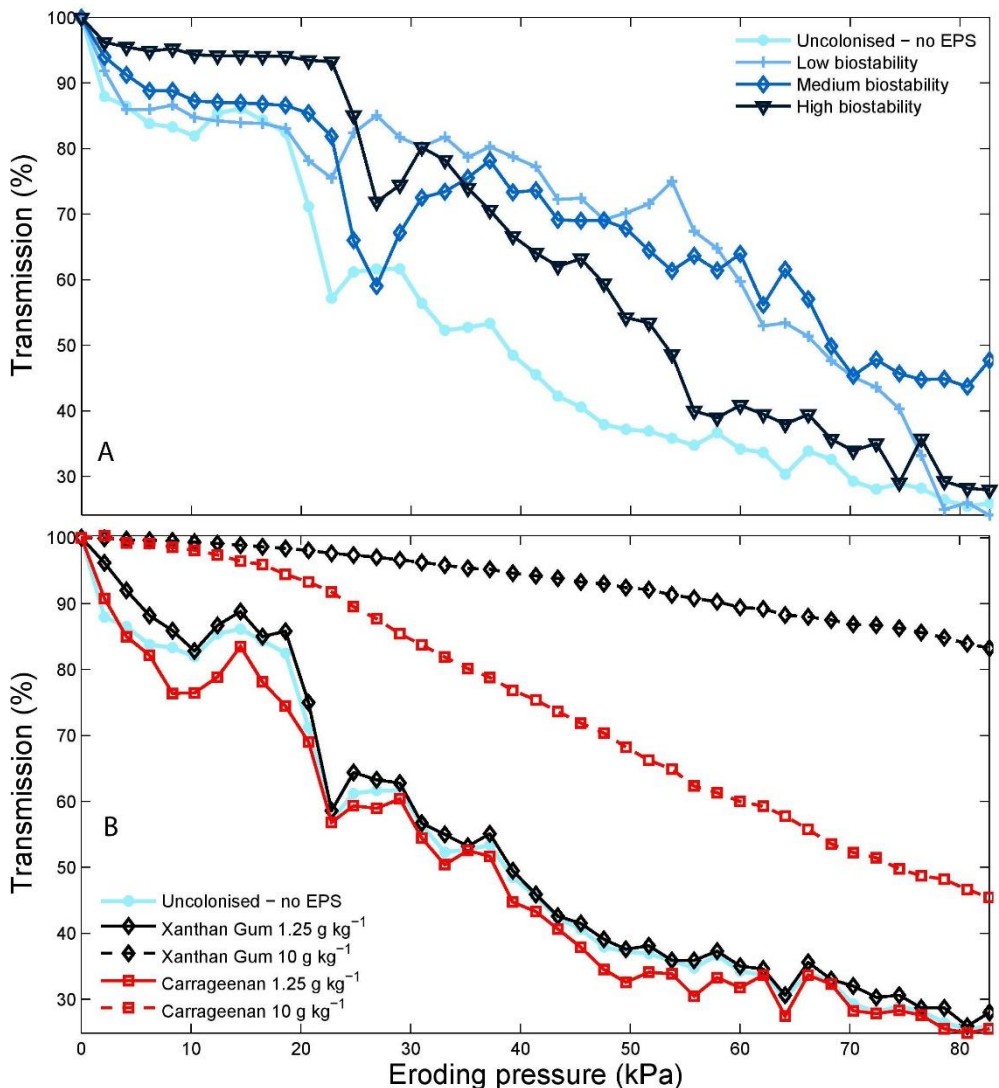

**Figure 10. CSM erosion profiles for sediment with different degrees of biostability due to natural biofilm colonisation (A) and due to different Xanthan Gum and Carrageenan synthetic EPS contents (B).**

Erosion profiles for low concentrations of synthetic Xanthan Gum and Carrageenan are similar to those

measured from the natural biostabilised sediments (Figure 10). For higher concentrations of Carrageenan



and particularly Xanthan Gum, the erosion rate is reduced relative to the natural biostabilised samples. In contrast to the natural samples where EPS concentration decreases with depth (Taylor and Paterson 1998), the synthetic EPS were mixed homogenously with depth in this study. As a consequence, the erosion rate for high concentrations of synthetic EPS has been reduced more than would be found under natural

conditions. To overcome this and to better replicate natural biofilm-mediated erosion behaviour, it may be more appropriate to apply synthetic EPS only on the surface in future studies. This will result in the highest EPS concentrations at the sediment surface that decreases with depth depending on the porosity and saturation of the substrate.

The methodologies described herein for preparing engineered sediments and the resultant biostabilisation

may serve as protocols to guide the design of future studies that aim to represent biological cohesion. In essence, biostabilisation effects of Xanthan Gum and Carrageenan synthetic EPS behave linearly (Figure 4) and are therefore predictable. Different concentrations of these synthetic EPS may be used to replicate the temporal and spatial variations generally seen in biostabilisation due to natural biofilm colonisation.. Other than biostabilisation, no differences in application or behaviour between Xanthan Gum and

Carrageen were observed in this study. Furthermore, the sensitivity analysis performed in this study showed that the effectiveness of Xanthan Gum and Carrageenan for the stabilisation of sediment, not only depends on the applied concentration, but is also is sensitive to the preparation procedure, time after application and environmental conditions. The results for the time elapsed after initial application tests were obtained for samples that dried out between measurements. Temporal behaviour of synthetic EPS

may be different when the engineered sediments remain wet for the duration of the test, which requires further research. The sensitivity of engineered sediments to salinity, pH and temperature found in this

study indicates that a high level of control of these environmental variables is required for reliable application of synthetic EPS in flume facilities.

Physical modelling of the complex flow, sediment transport and ecological interactions within aquatic
ecosystems is key to bridge the divide between field observations and numerical models (Thomas et al. 2014; Gerbersdorf and Wieprecht 2015). The implementation of biological processes into sediment transport equations that have traditionally been modelled as abiotic systems is expected to result in better predictions of sediment dynamics (Black et al. 2002; Righetti and Lucarelli 2007; Gerbersdorf et al. 2011; Parsons et al. 2016). Our study confirms that Xanthan Gum and Carrageenan synthetic EPS are not perfect
analogues of natural biofilms (Perkins et al. 2004), but they are capable of introducing realistic biological cohesion into flume facilities in a fast and controlled manner for a range of commonly used conditions. The reduction in experimental time here is significant since the maximum biostabilisation effects of natural biofilm can easily take 5 weeks or more to achieve, whereas synthetic EPS can be introduced at the same time as the sediment minimising time to set-up an experiment. Similarly growth patterns,
particularly the effect of increasing biostabilisation can easily be reproduced in a stepwise manner by introducing greater concentrations of the synthetic EPS. Although this study has focused on replicating one aspect of natural biofilm behaviour only, future physical modelling studies employing synthetic EPS may provide important insights into the role of biological cohesion in sediment dynamics, and how these may be altered in a changing climate.

## 5 Conclusions

This study aimed to evaluate biostabilisation effects of existing synthetic EPS for a range of conditions commonly used in physical modelling experiments. Four synthetic EPS were tested and addition of Xanthan Gum and Carrageenan increased the erosion threshold, while the addition of Alginic Acid and

Agar did not increase the erosion threshold for all test conditions. Changes in erosion thresholds produced by the addition of Xanthan Gum and Carrageenan synthetic EPS compared well to measured erosion threshold resulting from natural biofilm colonisation of the same sandy substrate. The increase of the erosion threshold with EPS content is linear and predictable for Xanthan Gum and Carrageenan, albeit with a lower rate of increase for Carrageenan. Furthermore, the effectiveness of Xanthan Gum and

Carrageenan to stabilise sediment is sensitive to the preparation procedure, time after application and environmental conditions such as salinity, pH and temperature. The methodologies for preparing engineered sediments described in this paper can provide quantifiable biostabilisation effects and may be used as protocols for designing future bio-physical experimental models that seek to represent biological cohesion. This approach will bring the significant advantages of being fast, replicable and controllable

which will improve experimental efficiency and enable experiments that explore a larger parameter space to be undertaken at lower cost.

**Acknowledgements.** WL, SJM and DRP were supported by the European Community's Horizon 2020 Programme through the grant to the budget of the Integrated Infrastructure Initiative HYDRALAB+,

Contract no. 654110. We thank Steve Darby and Chris Hackney from the University of Southampton for the generous loan of the CSM; Jane Reed for species ecology determination; Laura Jordan, Bas Bodewes

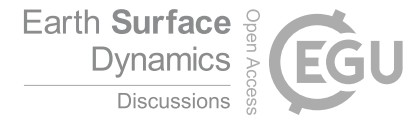

and Robert Houseago for their help with laboratory work and Brendan Murphy and Mark Anderson for technical support. The authors contributed in the following proportions to conception and design, data collection, analysis and conclusions, and manuscript preparation: WL (40%, 80%, 70%, 70%), SJM (40%, 20%, 20%, 20%) and DRP (20%, 0%, 10%, 10%).

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
