# Peer review of "Quantifying biostabilisation effects of biofilm-secreted and extracted extracellular polymeric substances (EPS) on sandy substrate"

_Earth Surface Dynamics, 2017_

## Referee Comment (RC1) · T. Tolhurst (Referee) · 14 Nov 2017

General comments: This paper presents the extensive testing of the effects of a diatomaceous biofilm and 4 different types of EPS at different concentrations and under different conditions of temperature, salinity and pH over different time scales and with different mixing procedures on the erosion threshold of a sandy sediment. It is clear and well written and provides novel insights into the contribution of EPS to biostabilisation.

[Figure]

Specific comments: Synthetic is used to describe the EPS used in this study, but it is not really synthetic – all of them have been extracted from natural sources, could a more accurate term be used? Page 2, line 12: I don't put a hyphen in microphytobenthos. I would not consider flocs and aggregates to be biofilms because to me a biofilm is a thin layer over a surface (although flocs and aggregates could have a biofilm over their surface). Page 3 lines 2-3: the terms 'microbial mats' and 'biofilms' are often used interchangeably, the former is not exclusively used to denote a covering of underlying sediments, and the latter is not exclusively used to denote coatings of single grains. In my own work, I use biofilm to denote a visible (either by eye or microscopically) layer of microphytobenthos on a sediment surface. Page 3 lines 10-13: there are also examples of buoyant biofilms, which reduce the erosion threshold of sediments (e.g. Sutherland, T. F., C. L. Amos, and J. Grant. "The effect of buoyant biofilms on the erodibility of sublittoral sediments of a temperate microtidal estuary." Limnology and Oceanography 43.2 (1998): 225-235; and Tolhurst, T. J., M. Consalvey, and D. M. Paterson. "Changes in cohesive sediment properties associated with the growth of a diatom biofilm." Hydrobiologia 596.1 (2008): 225-239). Page 5 line 14: to clearly differentiate from the synthetic EPS, I would insert 'diatom' before 'biofilm-secreted'. Page 7 line 20 and throughout: change 'Soil' to 'Sediment'. For me the sand used in this work is not a soil. Page 9 line 20: being precise, test Sand 7 increments in 2.068 kPa steps, but this probably doesn't matter too much given the error in the actual pressure of the CSM jet. Page 12 line 20: I'm not entirely sure what is meant by 'floated around the substrate' do you mean the diatoms were motile and not attached to the sediment grains? Page 15 lines 9-10: I'm not entirely sure what is meant by 'Added', can the authors clarify? Page 24 line 6: this reads oddly 'non-room temperature test conditions of 20°' isn't 20° room temperature? Should this be conditions of 10 and 40°? Page 25 Table 2: 'Relative biostabilisation' was termed 'biostabilisation index' by Manzenrieder, consider using this terminolgly instead (Manzenrieder, H. "Retardation of initial erosion under biological effects in sandy tidal flats." 1985 Australasian Conference on Coastal and Ocean Engineering. Institution of Engineers, Australia, 1985). Page 28

lines 18-19: This is interesting. I looked at the effects of letting diatom biofilms grown on sand drain and 'dry' out for a few hours in my PhD. There were changes in the erosion threshold and some indication that drier samples had a lower erosion threshold, but the effects were largely masked by variability in the biofilms. It is quite possible that the decrease in erosion threshold seen with time in this study is at least partly due to the successive drying. It seems quite likely to me that as EPS dries out it will become less effective at stabilising sediment, but as you say, it needs more research.

Technical corrections: Page 8 line 9: change 'weighted' to weighed'. Page 15 line 10: the 'Added' on line 10 should have a lower case A. Page 21 line 10: insert a comma after 'Gum'. Page 24 line 14: change 'linear' to 'linearly'. Page 28 line 13: delete second full stop. Page 30 line 14: insert a comma after 'controllable'.

---

## Referee Comment (RC2) · Anonymous Referee #2 · 20 Nov 2017

The paper examines the difference in the biostabilisation effects of natural biofilm colonisation and synthetic extracellular polymeric substances (EPS) in sand using a series of controlled flume experiments. Using an erosion meter, the critical shares stress required to entrainment clean and colonised sand are derived, and the effect of the preparation procedure for synthetic EPS on these values is investigated. The paper will be of great interest to those working on sediment transport and morphological changes in coastal and freshwater systems. The paper is worthy of publication but I would suggest the authors first explain the novelty of their work, improve the presen-

tation of the methods, and consider their confidence in the estimates of critical shear stress. Further details can be found below:

1. The authors provide a very useful review of the literature. However, having done so, I am left wondering what we do not understand, and thus why another study is required? I suggest the authors explain the novelty of their work.

2. I was disappointed the Introduction and Methods section did not make it clear what type of freshwater system is investigated. Which system are the scaled flume experiments trying to represent? I think this is especially important because we are told that one of the motivations for this study is that there has been a lot of work on biostabilisation in coastal settings but not in freshwater systems, and yet the studies biofilms are common in coastal zones. Furthermore, how do the studied conditions (e.g. slopes, depth:width, relative roughness, grain size, Reynolds number) pertain to those found in the natural system and match the conditions commonly found where these biofilms grow? Likewise, the authors should comment on how closely the Cohesive Strength Meter systems mimic erosion processes in the natural system? Furthermore little detail is provided on the setup of the small-scale synthetic EPS experiments. For example, I have read over the paper several times and I still cannot establish whether these tests were performed in a flume.

3. I authors state that synthetic EPS is able "to replicate the sediment stabilising capacity of natural biofilms". However the authors have found that three times more synthetic EPS concentration is needed to replicate the same stabilising effect of natural biofilms, suggesting the capacity is much higher for natural biofilms.

4. The calibration curve in equation 1 is important for gaining an accurate estimate of the critical shear stress. To allow readers to have confidence in their estimates, the authors should present a graph showing how this curve has been derived, and the predictive performance of this curve. Small deviations from the curve are likely to produce larger discrepancies in critical shear stress estimates due to the non-linear

relationship between critical shear stress and the applied jet force. For example, Figure 4 has error bars to represent the range in estimates from repeats, but hypothetically speaking, how much large would the error bar be if the uncertainty in the estimates themselves was incorporated?

Minor amendments:

1. P4, lines 14-21: There appears to be mismatch between this paragraph and the approach/results. If the prediction of the potential impacts of climate change on aquatic environments and the application of bioengineering adaptation strategies is important, how does this paper address these needs?

2. P9, line 16: What is routine S7?

3. Inconsistencies in the use of et al and author names in citations should be corrected.

---

## Editor Comment (EC1) · R. Hodge (Editor) · 1 Dec 2017

Both reviewers agree that this is an interesting paper that is worthy of publication following some revisions. Reviewer 1 identifies a number of points that require clarification, but which should be fairly straightforward to address. Reviewer 2 presents a slightly more critical appraisal of the paper. Some of their reservations could be addressed by clarifying the aim and novelty of the paper. The most novel aspect of this paper seems to be the systematic analysis of the impact of different types of synthetic EPS, however in the methods these experiments seem to be of secondary importance. In line

with Reviewer 2's comments, it would be helpful early on to explain the connections be-tween the two different sets of experiments (flumes and synthetic EPS). (Does it matter that the comparison between the flume and synthetic EPS experiments is comparing a sample from a flume bed with a far smaller sample in a petri dish?) You also need to clarify the novelty of the flume experiments over existing biofilm experiments. I don't think that it necessarily matters whether a specific system is being reproduced in the flume, but the choice of flume conditions should be justified. Both reviewers provide a number of minor comments that should also be taken into consideration. I look forward to receiving the revised version of this paper.

---

## Author Comment (AC1) · 20 Dec 2017

*Specific comments:*
*Synthetic is used to describe the EPS used in this study, but it is not really synthetic – all of them have been extracted from natural sources, could a more accurate term be used?*

We changed 'synthetic' to 'extracted' throughout the manuscript in an effort come up with a more accurate term.

*Page 2, line 12: I don't put a hyphen in microphytobenthos. I would not consider flocs and aggregates to be biofilms because to me a biofilm is a thin layer over a surface (although flocs and aggregates could have a biofilm over their surface).*

We removed the hyphen in microphytobenthos and deleted flocs and aggregates.

*Page 3 lines 2-3: the terms 'microbial mats' and 'biofilms' are often used interchangeably, the former is not exclusively used to denote a covering of underlying sediments, and the latter is not exclusively used to denote coatings of single grains. In my own work, I use biofilm to denote a visible (either by eye or microscopically) layer of microphytobenthos on a sediment surface.*

Thank you for pointing this out. We added this clarification to the manuscript (P3, lines 5 -7).

*Page 3 lines 10-13: there are also examples of buoyant biofilms, which reduce the erosion threshold of sediments (e.g. Sutherland, T. F., C. L. Amos, and J. Grant. "The effect of buoyant biofilms on the erodibility of sublittoral sediments of a temperate microtidal estuary." Limnology and Oceanography 43.2 (1998): 225-235; and Tolhurst, T. J., M. Consalvey, and D. M. Paterson. "Changes in cohesive sediment properties associated with the growth of a diatom biofilm." Hydrobiologia 596.1 (2008): 225-239).*

We added this information and the associated references to the manuscript (P3, lines 15 -17).

*Page 5 line 14: to clearly differentiate from the synthetic EPS, I would insert 'diatom' before 'biofilm-secreted'.*

Done

*Page 7 line 20 and throughout: change 'Soil' to 'Sediment'. For me the sand used in this work is not a soil.*

Done

*Page 9 line 20: being precise, test Sand 7 increments in 2.068 kPa steps, but this probably doesn't matter too much given the error in the actual pressure of the CSM jet.*

We corrected this information.

*Page 12 line 20: I'm not entirely sure what is meant by 'floated around the substrate' do you mean the diatoms were motile and not attached to the sediment grains?*

Yes indeed, that is what we meant to say. We rephrased to clarify the explanation, following the reviewer's suggestion (P13, lines 12-13).

*Page 15 lines 9-10: I'm not entirely sure what is meant by 'Added', can the authors clarify?*

We removed 'added' here and also in section heading 3.2.

*Page 24 line 6: this reads oddly 'non-room temperature test conditions of 20◦' isn't 20◦ room temperature? Should this be conditions of 10 and 40◦?*

We rephrased this sentence to correctly represent the temperature conditions (P25, lines 5-7).

*Page 25 Table 2: 'Relative biostabilisation' was termed 'biostabilisation index' by Manzenreider, consider using this terminology instead (Manzenrieder, H. "Retardation of initial erosion under biological effects in sandy tidal flats." 1985 Australasian Conference on Coastal and Ocean Engineering. Institution of Engineers, Australia, 1985).*

We changed 'relative biostabilisation' to 'biostabilisation index' in Table 2 in incorporated the citation to the work of Manzenreider (1985).

*Page 28 lines 18-19: This is interesting. I looked at the effects of letting diatom biofilms grown on sand drain and 'dry' out for a few hours in my PhD. There were changes in the erosion threshold and some indication that drier samples had a lower erosion threshold, but the effects were largely masked by variability in the biofilms. It is quite possible that the decrease in erosion threshold seen with time in this study is at least partly due to the successive drying. It seems quite likely to me that as EPS dries out it will become less effective at stabilising sediment, but as you say, it needs more research.*

Thank you for sharing your experiences on this topic. In our study, the engineered samples with repeat measurements over time showed different behavior after re-wetting the sediment. This may be due to a dilution effect of the EPS, the successive breakdown of the EPS over time, or some other unknown process associated with the successive drying. It would be insightful to further investigate this topic in future work.

*Technical corrections:*
*Page 8 line 9: change 'weighted' to weighed'.*
*Page 15 line 10: the 'Added' on line 10 should have a lower case A.*
*Page 21 line 10: insert a comma after 'Gum'.*
*Page 24 line 14: change 'linear' to 'linearly'.*

*Page 28 line 13: delete second full stop.*
*Page 30 line 14: insert a comma after 'controllable'.*

All done.

---

## Author Comment (AC2) · 20 Dec 2017

*1. The authors provide a very useful review of the literature. However, having done so, I am left wondering what we do not understand, and thus why another study is required? I suggest the authors explain the novelty of their work.*

Thank you for reviewing our work and your appreciation of the literature review. The development of a robust methodology and protocol for the application and impacts of extracted EPS in flume facilities provides the novelty of our work (this is explicitly mentioned and has been given emphasis on P5, lines 14-17). Indeed, earlier studies have investigated natural diatom-biofilm behaviour (e.g. Gerbersdorf and Wieprecht, 2015) and also work was done on extracted EPS already (e.g. Tolhurst 2002). Our study builds on this work and explicitly relates the sediment stabilising ability of natural diatom-biofilms to that of extracted EPS. A unique aspect of our study is that we use the same sediment for the natural diatom-biofilms and extracted EPS tests so we can compare the results directly (this is explicitly mentioned on P6, line 1 and P6, lines 11-13). In addition, we expand the existing knowledge on the application and impacts of extracted EPS by testing four different EPS for a range of environmental conditions. Such knowledge is currently lacking and has led to costly and time-consuming trial-and-error approaches in a variety of different modelling facilities. Our findings present a systematic methodology and protocol for a range of commercially available EPS and are therefore expected to inform future studies seeking to introduce biological cohesion in a rapid and controlled manner (the importance for preparation time and experimental duration are emphasised on P5 line 6-9).

*2. I was disappointed the Introduction and Methods section did not make it clear what type of freshwater system is investigated. Which system are the scaled flume experiments trying to represent? I think this is especially important because we are told that one of the motivations for this study is that there has been a lot of work on biostabilisation in coastal settings but not in freshwater systems, and yet the studies biofilms are common in coastal zones. Furthermore, how do the studied conditions (e.g. slopes, depth:width, relative roughness, grain size, Reynolds number) pertain to those found in the natural system and match the conditions commonly found where these biofilms grow? Likewise, the authors should comment on how closely the Cohesive Strength Meter systems mimic erosion processes in the natural system? Furthermore little detail is provided on the setup of the small-scale synthetic EPS experiments. For example, I have read over the paper several times and I still cannot establish whether these tests were performed in a flume.*

We use brackish water in our experiment. This experimental condition is explicitly mentioned on P7, lines 15-16. The brackish water setting is representative of estuarine, mangrove and deltaic settings within the fluvial-to-marine transition zone. In our literature review, we indeed mention that the role of EPS in freshwater systems is not as well understood as in marine systems (P2, lines 16-20) but our experiment was not aimed at gaining a better understanding of the EPS behaviour in freshwater conditions.

The experimental design conditions are in approximate agreement with natural reference systems. Please note that the experiment was not setup to replicate a specific natural system but rather a collection of shallow brackish environments. The channels had no initial gradient (but the flow may have

created a self-formed gradient in the substrate during the experiment), a width-to-depth ratio of 5, 110 microns sand and a Reynolds number indicating turbulent flow (Re = 5000 – 10000). We added this information to the Methods section 2.1.1. But most importantly, these experimental conditions resulted in a thriving biofilm with a species composition that was consistent with species commonly seen in brackish coastal environments (P13, lines 6-15).

The Cohesive Strength Meter employs a vertical jet to measure the erosion shear stress of sediments. This approach differs from natural erosion processes, which predominantly generate a horizontal shear. Based on a series of systematic tests, Tolhurst et al (1999) provides a calibration of the vertical jet to an equivalent critical erosion shear stress. A full discussion on the strengths and weaknesses of the CSM erosion device as well as the development history and relation to other erosion devices is provided in Tolhurst et al (1999) and we refer the reviewer to this document for full details. In our study, we applied the calibration of the vertical jet to an equivalent critical erosion shear stress, and we would also like to stress that the CSM provides one of the few erosion devices allowing workers to make quantitative and repeat measurements of sediment stability.

The small-scale tests with extracted EPS are performed in petri dishes. This is for example explicitly written on P11, line 11-12 ('The sand-EPS mixture was then poured into plastic petri dishes') and lines 12-13 ('therefore care was taken to create a level surface by tapping the side of the petri dishes before testing'). We also refer to the protocol used in Tolhurst et al (2002) and mention that we follow a similar protocol. To make it more explicit that these small-scale tests were performed in petri dishes, we added this information to section heading 3.1 (Petri dish sediment sample tests with extracted EPS) as well as in referring to the protocol used in Tolhurst et al (2002) on P11, lines 5-6.

*3. authors state that synthetic EPS is able "to replicate the sediment stabilising capacity of natural biofilms". However the authors have found that three times more synthetic EPS concentration is needed to replicate the same stabilising effect of natural biofilms, suggesting the capacity is much higher for natural biofilms.*

Our findings indeed indicate that extracted EPS can replicate the sediment stabilising capacity of natural biofilms as seen from the biostabilisation index (Table 2). In contrast to the reviewer's suggestion, we do not think that the natural biofilms have a higher stabilizing capacity than observed in our study. The biostabilisation index values are consistent with earlier studies on the sediment stabilising capacity of natural biofilms (Paterson 1989; Dade et al. 1990; Amos et al. 1998; Tolhurst et al. 1999; Tolhurst et al. 2003; Friend et al. 2003; Friend, Collins, and Holligan 2003; Droppo et al. 2007; Righetti and Lucarelli 2007; Vignaga, Haynes, and Sloan 2012; Graba et al. 2013; Thom et al. 2015). It is therefore unlikely that the capacity is much higher for natural biofilms, although the observations in our study indicate a substantial internal variation (Figure 2). Rather, the explanation for the different EPS concentration in the extracted EPS tests and natural biofilm experiment must be sought in the determination of the EPS concentration in the natural biofilm experiment. We provide two explanations for the lower EPS concentrations in the biofilm experiment (P27, lines 6 to P28, line 7) and both may explain the difference with the applied extracted EPS concentrations in the small-scale petri dish tests.

*4. The calibration curve in equation 1 is important for gaining an accurate estimate of the critical shear stress. To allow readers to have confidence in their estimates, the authors should present a graph showing how this curve has been derived, and the predictive performance of this curve. Small deviations from the curve are likely to produce larger discrepancies in critical shear stress estimates due to the non-linear relationship between critical shear stress and the applied jet force. For example, Figure 4 has error bars to represent the range in estimates from repeats, but hypothetically speaking, how much large would the error bar be if the uncertainty in the estimates themselves was incorporated?*

We refer to Tolhurst et al (2002) for details on the how the calibration curve is derived. This article provides a detailed explanation of the performed tests and the quality of the calibration curve. As shown in Tolhurst et al (2002), the uncertainty in the calibration curve is typically in the order of 0.1-0.2 $N/m^2$, which suggests that the error bars would be 0.4 $N/m^2$ when, hypothetically, taking this effect into account.

*Minor amendments:*

*P4, lines 14-21: There appears to be mismatch between this paragraph and the approach/results. If the prediction of the potential impacts of climate change on aquatic environments and the application of bioengineering adaptation strategies is important, how does this paper address these needs?*

The sentences on P4 (lines 18-21) and P5 (lines 1-9) are included to signal the need for including biological processes in sediment transport predictions. The understanding of these bio-physical relations is currently limited and the relationships may also be different under different climatic conditions. We consider flume experiments a primary tool of researchers to address these bio-physical relationships but the developments are hampered because experiments including real biota are time-consuming and costly, with also some questions raised about the degree of natural behaviour of biota in flume facilities.

Our paper provides the first step to overcome the aforementioned issues by the development of a robust methodology and protocol for the application and resultant impacts of extracted EPS to introduce biological cohesion in a rapid and controlled manner. So although the current paper does not directly address the potential impacts of climate change on aquatic environments, the work described provides an important step in facilitating future work that will do so. We believe that providing this context is helpful to the reader and have checked the manuscript to make sure that the implications of our work are well represented, also with reference to studying the impacts of climate change.

*P9, line 16: What is routine S7?*

Routine S7 is one of the CSM test routines. To further clarify this, we now explicitly link the use of S7 in the manuscript to the word 'CSM'. See P10, lines 7-12.

*Inconsistencies in the use of et al and author names in citations should be corrected.*

Done.

---

## Author Comment (AC3) · 20 Dec 2017

Dear Dr. Hodge,

Please find enclosed the revised (Original MS Reference esurf-2017-59) manuscript entitled *Quantifying biostabilisation effects of biofilm-secreted and synthetic extracellular polymeric substances (EPS) on sandy substrate* by W.I. van de Lageweg, S.J. McLelland and D.R. Parsons.

We found the reviews helpful and constructive. Below we describe how we used the reviews (in italics) to improve the manuscript. Detailed textual comments were mostly incorporated and sometimes used as indicator where text clarity had to be improved.

Thank you for your guidance in revising the manuscript. The development of a robust methodology and protocol for the application and impacts of extracted EPS in flume facilities provides the novelty of our work. This novel aspect is explicitly mentioned and has been given emphasis on P5, lines 14-17. We now also more clearly outline the connections between the experiments and the motivation for the petri-dish tests (the importance for preparation time and experimental duration are emphasised on P5 line 7-9). Furthermore, we clarify what type of natural system our flume experiments are intended to reproduce (P7, lines 15-16) and we show in the findings that the biofilm species composition is in agreement with species commonly seen in these natural environments (P13, lines 6-15). Lastly, we provide more details on the flume experiment design conditions (Methods section 2.1.1).

Sincerely,
Wietse van de Lageweg, on behalf of all authors

*Specific comments:*
*Synthetic is used to describe the EPS used in this study, but it is not really synthetic – all of them have been extracted from natural sources, could a more accurate term be used?*

We changed 'synthetic' to 'extracted' throughout the manuscript in an effort come up with a more accurate term.

*Page 2, line 12: I don't put a hyphen in microphytobenthos. I would not consider flocs and aggregates to be biofilms because to me a biofilm is a thin layer over a surface (although flocs and aggregates could have a biofilm over their surface).*

We removed the hyphen in microphytobenthos and deleted flocs and aggregates.

*Page 3 lines 2-3: the terms 'microbial mats' and 'biofilms' are often used interchangeably, the former is not exclusively used to denote a covering of underlying sediments, and the latter is not exclusively used to denote coatings of single grains. In my own work, I use biofilm to denote a visible (either by eye or microscopically) layer of microphytobenthos on a sediment surface.*

Thank you for pointing this out. We added this clarification to the manuscript (P3, lines 5 -7).

*Page 3 lines 10-13: there are also examples of buoyant biofilms, which reduce the erosion threshold of sediments (e.g. Sutherland, T. F., C. L. Amos, and J. Grant. "The effect of buoyant biofilms on the erodibility of sublittoral sediments of a temperate microtidal estuary." Limnology and Oceanography 43.2 (1998): 225-235; and Tolhurst, T. J., M. Consalvey, and D. M. Paterson. "Changes in cohesive sediment properties associated with the growth of a diatom biofilm." Hydrobiologia 596.1 (2008): 225-239).*

We added this information and the associated references to the manuscript (P3, lines 15 -17).

*Page 5 line 14: to clearly differentiate from the synthetic EPS, I would insert 'diatom' before 'biofilm-secreted'.*

Done

*Page 7 line 20 and throughout: change 'Soil' to 'Sediment'. For me the sand used in this work is not a soil.*

Done

*Page 9 line 20: being precise, test Sand 7 increments in 2.068 kPa steps, but this probably doesn't matter too much given the error in the actual pressure of the CSM jet.*

We corrected this information.

*Page 12 line 20: I'm not entirely sure what is meant by 'floated around the substrate' do you mean the diatoms were motile and not attached to the sediment grains?*

Yes indeed, that is what we meant to say. We rephrased to clarify the explanation, following the reviewer's suggestion (P13, lines 12-13).

*Page 15 lines 9-10: I'm not entirely sure what is meant by 'Added', can the authors clarify?*

We removed 'added' here and also in section heading 3.2.

*Page 24 line 6: this reads oddly 'non-room temperature test conditions of 20◦' isn't 20◦ room temperature? Should this be conditions of 10 and 40◦?*

We rephrased this sentence to correctly represent the temperature conditions (P25, lines 5-7).

*Page 25 Table 2: 'Relative biostabilisation' was termed 'biostabilisation index' by Manzenreider, consider using this terminologly instead (Manzenrieder, H. "Retardation of initial erosion under biological effects in sandy tidal flats." 1985 Australasian Conference on Coastal and Ocean Engineering. Institution of Engineers, Australia, 1985).*

We changed 'relative biostabilisation' to 'biostabilisation index' in Table 2 in incorporated the citation to the work of Manzenreider (1985).

*Page 28 lines 18-19: This is interesting. I looked at the effects of letting diatom biofilms grown on sand drain and 'dry' out for a few hours in my PhD. There were changes in the erosion threshold and some indication that drier samples had a lower erosion threshold, but the effects were largely masked by variability in the biofilms. It is quite possible that the decrease in erosion threshold seen with time in this study is at least partly due to the successive drying. It seems quite likely to me that as EPS dries out it will become less effective at stabilising sediment, but as you say, it needs more research.*

Thank you for sharing your experiences on this topic. In our study, the engineered samples with repeat measurements over time showed different behavior after re-wetting the sediment. This may be due to a dilution effect of the EPS, the successive breakdown of the EPS over time, or some other unknown process associated with the successive drying. It would be insightful to further investigate this topic in future work.

*Technical corrections:*
*Page 8 line 9: change 'weighted' to weighed'.*
*Page 15 line 10: the 'Added' on line 10 should have a lower case A.*
*Page 21 line 10: insert a comma after 'Gum'.*
*Page 24 line 14: change 'linear' to 'linearly'.*

*Page 28 line 13: delete second full stop.*
*Page 30 line 14: insert a comma after 'controllable'.*

All done.

**Reviewer 2**

*1. The authors provide a very useful review of the literature. However, having done so, I am left wondering what we do not understand, and thus why another study is required? I suggest the authors explain the novelty of their work.*

Thank you for reviewing our work and your appreciation of the literature review. The development of a robust methodology and protocol for the application and impacts of extracted EPS in flume facilities provides the novelty of our work (this is explicitly mentioned and has been given emphasis on P5, lines 14-17). Indeed, earlier studies have investigated natural diatom-biofilm behaviour (e.g. Gerbersdorf and Wieprecht, 2015) and also work was done on extracted EPS already (e.g. Tolhurst 2002). Our study builds on this work and explicitly relates the sediment stabilising ability of natural diatom-biofilms to that of extracted EPS. A unique aspect of our study is that we use the same sediment for the natural diatom-biofilms and extracted EPS tests so we can compare the results directly (this is explicitly mentioned on P6, line 1 and P6, lines 11-13). In addition, we expand the existing knowledge on the application and impacts of extracted EPS by testing four different EPS for a range of environmental conditions. Such knowledge is currently lacking and has led to costly and time-consuming trial-and-error approaches in a variety of different modelling facilities. Our findings present a systematic methodology and protocol for a range of commercially available EPS and are therefore expected to inform future studies seeking to introduce biological cohesion in a rapid and controlled manner (the importance for preparation time and experimental duration are emphasised on P5 line 6-9).

*2. I was disappointed the Introduction and Methods section did not make it clear what type of freshwater system is investigated. Which system are the scaled flume experiments trying to represent? I think this is especially important because we are told that one of the motivations for this study is that there has been a lot of work on biostabilisation in coastal settings but not in freshwater systems, and yet the studies biofilms are common in coastal zones. Furthermore, how do the studied conditions (e.g. slopes, depth:width, relative roughness, grain size, Reynolds number) pertain to those found in the natural system and match the conditions commonly found where these biofilms grow? Likewise, the authors should comment on how closely the Cohesive Strength Meter systems mimic erosion processes in the natural system? Furthermore little detail is provided on the setup of the small-scale synthetic EPS experiments. For example, I have read over the paper several times and I still cannot establish whether these tests were performed in a flume.*

We use brackish water in our experiment. This experimental condition is explicitly mentioned on P7, lines 15-16. The brackish water setting is representative of estuarine, mangrove and deltaic settings within the fluvial-to-marine transition zone. In our literature review, we indeed mention that the role of EPS in freshwater systems is not as well understood as in marine systems (P2, lines 16-20) but our experiment was not aimed at gaining a better understanding of the EPS behaviour in freshwater conditions.

The experimental design conditions are in approximate agreement with natural reference systems. Please note that the experiment was not setup to replicate a specific natural system but rather a collection of shallow brackish environments. The channels had no initial gradient (but the flow may have

created a self-formed gradient in the substrate during the experiment), a width-to-depth ratio of 5, 110 microns sand and a Reynolds number indicating turbulent flow (Re = 5000 – 10000). We added this information to the Methods section 2.1.1. But most importantly, these experimental conditions resulted in a thriving biofilm with a species composition that was consistent with species commonly seen in brackish coastal environments (P13, lines 6-15).

The Cohesive Strength Meter employs a vertical jet to measure the erosion shear stress of sediments. This approach differs from natural erosion processes, which predominantly generate a horizontal shear. Based on a series of systematic tests, Tolhurst et al (1999) provides a calibration of the vertical jet to an equivalent critical erosion shear stress. A full discussion on the strengths and weaknesses of the CSM erosion device as well as the development history and relation to other erosion devices is provided in Tolhurst et al (1999) and we refer the reviewer to this document for full details. In our study, we applied the calibration of the vertical jet to an equivalent critical erosion shear stress, and we would also like to stress that the CSM provides one of the few erosion devices allowing workers to make quantitative and repeat measurements of sediment stability.

The small-scale tests with extracted EPS are performed in petri dishes. This is for example explicitly written on P11, line 11-12 ('The sand-EPS mixture was then poured into plastic petri dishes') and lines 12-13 ('therefore care was taken to create a level surface by tapping the side of the petri dishes before testing'). We also refer to the protocol used in Tolhurst et al (2002) and mention that we follow a similar protocol. To make it more explicit that these small-scale tests were performed in petri dishes, we added this information to section heading 3.1 (Petri dish sediment sample tests with extracted EPS) as well as in referring to the protocol used in Tolhurst et al (2002) on P11, lines 5-6.

*3. authors state that synthetic EPS is able "to replicate the sediment stabilising capacity of natural biofilms". However the authors have found that three times more synthetic EPS concentration is needed to replicate the same stabilising effect of natural biofilms, suggesting the capacity is much higher for natural biofilms*.

Our findings indeed indicate that extracted EPS can replicate the sediment stabilising capacity of natural biofilms as seen from the biostabilisation index (Table 2). In contrast to the reviewer's suggestion, we do not think that the natural biofilms have a higher stabilizing capacity than observed in our study. The biostabilisation index values are consistent with earlier studies on the sediment stabilising capacity of natural biofilms (Paterson 1989; Dade et al. 1990; Amos et al. 1998; Tolhurst et al. 1999; Tolhurst et al. 2003; Friend et al. 2003; Friend, Collins, and Holligan 2003; Droppo et al. 2007; Righetti and Lucarelli 2007; Vignaga, Haynes, and Sloan 2012; Graba et al. 2013; Thom et al. 2015). It is therefore unlikely that the capacity is much higher for natural biofilms, although the observations in our study indicate a substantial internal variation (Figure 2). Rather, the explanation for the different EPS concentration in the extracted EPS tests and natural biofilm experiment must be sought in the determination of the EPS concentration in the natural biofilm experiment. We provide two explanations for the lower EPS concentrations in the biofilm experiment (P27, lines 6 to P28, line 7) and both may explain the difference with the applied extracted EPS concentrations in the small-scale petri dish tests.

*4. The calibration curve in equation 1 is important for gaining an accurate estimate of the critical shear stress. To allow readers to have confidence in their estimates, the authors should present a graph showing how this curve has been derived, and the predictive performance of this curve. Small deviations from the curve are likely to produce larger discrepancies in critical shear stress estimates due to the non-linear relationship between critical shear stress and the applied jet force. For example, Figure 4 has error bars to represent the range in estimates from repeats, but hypothetically speaking, how much large would the error bar be if the uncertainty in the estimates themselves was incorporated?*

We refer to Tolhurst et al (2002) for details on the how the calibration curve is derived. This article provides a detailed explanation of the performed tests and the quality of the calibration curve. As shown in Tolhurst et al (2002), the uncertainty in the calibration curve is typically in the order of 0.1-0.2 $N/m^2$, which suggests that the error bars would be 0.4 $N/m^2$ when, hypothetically, taking this effect into account.

*Minor amendments:*

*P4, lines 14-21: There appears to be mismatch between this paragraph and the approach/results. If the prediction of the potential impacts of climate change on aquatic environments and the application of bioengineering adaptation strategies is important, how does this paper address these needs?*

The sentences on P4 (lines 18-21) and P5 (lines 1-9) are included to signal the need for including biological processes in sediment transport predictions. The understanding of these bio-physical relations is currently limited and the relationships may also be different under different climatic conditions. We consider flume experiments a primary tool of researchers to address these bio-physical relationships but the developments are hampered because experiments including real biota are time-consuming and costly, with also some questions raised about the degree of natural behaviour of biota in flume facilities.

Our paper provides the first step to overcome the aforementioned issues by the development of a robust methodology and protocol for the application and resultant impacts of extracted EPS to introduce biological cohesion in a rapid and controlled manner. So although the current paper does not directly address the potential impacts of climate change on aquatic environments, the work described provides an important step in facilitating future work that will do so. We believe that providing this context is helpful to the reader and have checked the manuscript to make sure that the implications of our work are well represented, also with reference to studying the impacts of climate change.

*P9, line 16: What is routine S7?*

Routine S7 is one of the CSM test routines. To further clarify this, we now explicitly link the use of S7 in the manuscript to the word 'CSM'. See P10, lines 7-12.

*Inconsistencies in the use of et al and author names in citations should be corrected.*

Done.

[revised manuscript text omitted]

Baptist. 2015. "Sustainable Hydraulic Engineering through Building with Nature." *Journal of Hydro-Environment Research* 9 (2): 159–71. https://doi.org/10.1016/j.jher.2014.06.004.

Zanke, U.C.E. 2003. "On the Influence of Turbulence on the Initiation of Sediment Motion." *International

Journal of Sediment Research* 18 (1): 1–15.

---

## Editor Decision (ED1)

Apologies for the delay in getting these comments to you. I think that you have done a good job of responding to the reviewers' comments, and that the paper is not far off being acceptable for publication. However, I am suggesting some minor revisions that would help to draw together the two different experimental components, and there are still some areas that would benefit from further clarification and signposting for the reader. I look forward to seeing the revised paper.

Page/line

1/17: Change 'provide' to 'provides'

1/18: Change to 'mean biostabilisation effect'

1/20: Clarify that the critical erosion threshold are of colonised sand

2/1: Measured range of what?

3/11: I think that three author papers should be 'et al' in the text, not written in full.

3/13: Consider rephrasing – for example 'sediment stability measured as the threshold for erosion'.

3/17: 'Yet' is not needed.

4/5: Clarify that you are referring to dynamics at the scale of the entire estuary.

4/16: Consider changing to 'sediment settling rates'.

5/4: At some point you need to clearly explain what you mean by extracted EPS – this could be here, or in line 4/11.

5/11: Somewhere in this paragraph you need to mention the natural biofilm experiments, e.g. by saying that you will be comparing the extracted EPS results to natural biofilms in complementary experiments. Otherwise the natural biofilms are not mentioned until aim 1, and then it's not clear where they have come from.

6/10: This paragraph also needs some restructuring for clarity. Be explicit that there are two sets of experiments being reported; one using natural biofilms, and the other using extracted EPS. For example: 'In the first set of experiments… In the second set…' Don't use the word 'auxiliary'; this means supplementary, whereas you have identified the extracted EPS experiments as being the main aim of this research. I would also be consistent with how you label the sections headings so that there are clearly two sections of the methods and results.

7/4: Figures are placed near their first occurrence in the text. Consider moving Figure 1 to here, or removing this reference to it.

7/7: You don't refer to the 1 mm and mixed channels anywhere else in the paper, so no need to include them in the methods.

7/12: Replace 'auxiliary'

8/6: Consider rephrasing: the flume was inoculated using

8/12: 'From the top 0.01 m' could be interpreted as being from the furthest upstream 0.01 m of the channel. Clarify that you are referring to 0.01 m in the vertical, and explain how the sampling sites were selected. Did you specifically target locations with visible biofilm, or sample from all channel locations?

8/13 and elsewhere: Where section titles have been inserted into the text there is a missing space afterwards. Remove the title and just leave Section 2.1.1 (in this case).

10/10: It's still not entirely clear how this works. Do you just increase the applied force until the surface is eroded? How do you decide when to stop? Figure 10 has a transmission %, but you don't explain anywhere what this is. How do you go from the datasets in Figure 10 to a single erosion threshold value?

10/14: 'Enabling' is a better word that 'allowing for'.

10/18: Start this paragraph with words or a sentence to tell the reader that you are now moving onto the second set of experiments.

12/15: The start of this line is awkwardly phrase; rephrase.

12/16: Use °C rather than writing out in full.

12/17: What liquids were used to give these pHs?

13/1: Add a subheading to explain that this section is referring to the natural biofilms experiment.

13/4: Delete 'a' before darker

14/14: How is this theoretical entrainment threshold determined? Delete 'applied'.

14/15: How many of these samples came from sections of the flume where no biofilm was present? Without knowing the sampling strategy, this isn't clear. You could use the EPS content values to say something about this.

15/3: There is clearly more in these data than you currently present, but I agree that these data are not the key point of the paper. However, as these data are compared to the extracted EPS data, it would be helpful to explore them a little more. Could you add the time data onto Fig 2, for example by colouring the points by week? How did variation over time compare to the spatial variation within and between the flumes at any given time? For all the extracted EPS data you plot shear stress against amount of EPS. Why not also plot the natural biofilm data like this?

16/6: Make it clear that this section is moving onto the extracted EPS experiments. For example, refer to the 'second set' of experiments (or whatever you call them), rather than just small scale experiments (a phrase that I don't think you've used before).

17/1: This paragraph mainly seems to repeat what you have already said in the methods. 17/5 onwards seems to be new information, which should be moved to the methods.

18/Fig. 4 and other figures: Add space between = and following number.

18/Table 1: How many replicates?

20/Fig 5: A good way of comparing the natural and extracted EPS shear stress values would be to add dashed lines for the mean (and median/standard deviation?) of the shear stresses measured for the natural biofilms. This applies to all figures like this (5/7/8/9).

Also, is mixing different to stirring?

22/1: Rephrase to clarify that you are referring to the extracted EPS experiments.

22/Fig 7: The caption refers to tap water, but distilled water was referred to at the end of 2.3.2.

26/5: Explain how a mean biostabilisation index is calculated.

26/13: Quote some figures to support this claim that the index is similar.

26/14: Change to 'more suited for replicating'

27/Table 2: The caption still doesn't really explain the biostabilisation index. This table could be clearer. It's a bit confusing that the top row of headings doesn't apply all the way down the columns. What statistic is quoted for the extracted EPS results (mean/median)?

27/4: Give some values.

27/7: I think that you're saying that this technique doesn't measure all of the EPS that is in the sample?

29/Fig 10: See earlier comment about explaining what transmission is.

30/17: Is the change over time primarily from the sediment drying, rather than any sort of degradation of the extracted EPS?

---

## Author Response (AR2)

Dear Dr. Hodge,

Please find enclosed the revised (MS Reference esurf-2017-59) manuscript entitled *Quantifying biostabilisation effects of biofilm-secreted and extracted extracellular polymeric substances (EPS) on sandy substrate* by W.I. van de Lageweg, S.J. McLelland and D.R. Parsons.

Thank you for your guidance in further improving the manuscript. We included your suggestions to more clearly outline the connections between the two experimental components and also added your suggestions in terms of signposting and clarification to more clearly communicate the results of our work to the reader. Please find a detailed point-by-point response to all your comments (*in italics*) below.

Sincerely,
Wietse van de Lageweg, on behalf of all authors

Page/line

*1/17: Change 'provide' to 'provides'*
Done

*1/18: Change to 'mean biostabilisation effect'*
Done

*1/20: Clarify that the critical erosion threshold are of colonised sand*
Done

*2/1: Measured range of what?*
Of critical erosion thresholds. We added this information to the main text.

*3/11: I think that three author papers should be 'et al' in the text, not written in full.*
We used Mendeley as our Reference Manager and selected a citation style in which three paper authors were written in full, and not as 'et al'. We checked the document and corrected all occurrences.

*3/13: Consider rephrasing – for example 'sediment stability measured as the threshold for erosion'.*
Done. Here and in other places throughout the manuscript.

*3/17: 'Yet' is not needed.*
We removed 'Yet'.

*4/5: Clarify that you are referring to dynamics at the scale of the entire estuary.*
Done

*4/16: Consider changing to 'sediment settling rates'.*
Done

*5/4: At some point you need to clearly explain what you mean by extracted EPS – this could be here, or in line 4/11.*
We added a definition of extracted EPS and information on its usage in the current study (P4, L18-21).

*5/11: Somewhere in this paragraph you need to mention the natural biofilm experiments, e.g. by saying that you will be comparing the extracted EPS results to natural biofilms in complementary experiments. Otherwise the natural biofilms are not mentioned until aim 1, and then it's not clear where they have come from.*
Done

*6/10: This paragraph also needs some restructuring for clarity. Be explicit that there are two sets of experiments being reported; one using natural biofilms, and the other using extracted EPS. For example: 'In the first set of experiments… In the second set…' Don't use the word 'auxiliary'; this means supplementary, whereas you have identified the extracted EPS experiments as being the main aim of this research. I would also be consistent with how you label the sections headings so that there are clearly two sections of the methods and results.*
As suggested, we have made it more explicit that we are reporting on two experiments. Both in the sentences leading up to the specific aims (P6, L3-6) and in listing the aims (P6, L13-17).

*7/4: Figures are placed near their first occurrence in the text. Consider moving Figure 1 to here, or removing this reference to it.*
We moved Figure 1 to the top of P8.

*7/7: You don't refer to the 1 mm and mixed channels anywhere else in the paper, so no need to include them in the methods.*
We removed the sentences about the four channels with other substrates not reported on in this study.

*7/12: Replace 'auxiliary'*
Done

*8/6: Consider rephrasing: the flume was inoculated using*
Done

*8/12: 'From the top 0.01 m' could be interpreted as being from the furthest upstream 0.01 m of the channel. Clarify that you are referring to 0.01 m in the vertical, and explain how the sampling sites were selected. Did you specifically target locations with visible biofilm, or sample from all channel locations?*
We clarified now that we are referring to the top 0.01 m in the vertical. Additionally, we clarified that we targeted locations with visible biofilm (P9, L12-15).

*8/13 and elsewhere: Where section titles have been inserted into the text there is a missing space afterwards. Remove the title and just leave Section 2.1.1 (in this case).*
Done

*10/10: It's still not entirely clear how this works. Do you just increase the applied force until the surface is eroded? How do you decide when to stop? Figure 10 has a transmission %, but you don't explain anywhere what this is. How do you go from the datasets in Figure 10 to a single erosion threshold value?*
The force of the jet pulse is increased depending on the programmed selected. In our study, we selected the pre-programmed S7 routine, which starts 0 kPa incrementing by 2.068 kPa per step up to 82.74 kPa with a jet being fired for 1 second. 36 other pre-programmed routines consist of other settings for the air increments, duration of the jet, and maximum air pressure applied.

The CSM device works through the program, independent of surface erosion. Therefore, all tests have an identical duration. If no erosion is observed, the maximum air pressure applied can be increased to possibly erode the surface at the next attempt. Initially, some trial-and-error tests are required to obtain the correct CSM routine for the studied material. We found that routine S7 provided a balance between sufficiently detailed measurements for the lower threshold of erosion of some samples and a large measurement range required for some of the engineered substrates (i.e. Xanthan Gum).

A drop in transmission indicates an erosion event. The erosion profile usually has three different parts:
1. An initial horizontal line where the transmission is close to 100%.
2. A slope representing the drop in transmission of light across the measurement chamber as erosion occurs and sediment is being suspended.
3. An asymptotic part where the transmission approaches 0 when the air pressure increases.

These profiles vary depending on the sediment properties. Following Tolhurst et al (1999), the critical erosion threshold was defined as the pressure step at which the transmission drops below 90%.

We added some of the above information to the main text and would like to refer readers to Tolhurst et al (1999) for a full description of the CSM device.

*10/14: 'Enabling' is a better word that 'allowing for'.*
Done

*10/18: Start this paragraph with words or a sentence to tell the reader that you are now moving onto the second set of experiments.*
Done

*12/15: The start of this line is awkwardly phrase; rephrase.*
Done

*12/16: Use °C rather than writing out in full.*
Done, here and in other places.

*12/17: What liquids were used to give these pHs?*
We used standard and commercially available buffer solutions to obtain liquids with these pHs. We added this information to the manuscript (P14, L17-18).

*13/1: Add a subheading to explain that this section is referring to the natural biofilms experiment.*
Done

*13/4: Delete 'a' before darker*
Done

*14/14: How is this theoretical entrainment threshold determined? Delete 'applied'.*
We calculated the theoretical entrainment threshold $\tau_c$ for our sediment according:
$$\tau_c = \theta_c \cdot (\rho_s - \rho) \cdot g \cdot D_{50}$$
where $\theta_c$ is the Shields number (N/m$^2$), $\rho_s$ is density of sediment (kg/m$^3$), $\rho$ is the density of water (kg/m$^3$) and $D_{50}$ is the median grain size (m).

The Shields number $\theta_c$ is calculated following Zanke (2003):
$$\theta_c = 0.145 \cdot \mathrm{Re}_p^{-0.33} + 0.045 \cdot 10^{-1100 \cdot \mathrm{Re}_p^{-1.5}}$$
where $\mathrm{Re}_p$ is the Reynolds particle number and calculated by:

$$\mathrm{Re}_p = D_{50}^{1.5} \cdot \frac{\sqrt{\Delta \cdot g}}{\nu}$$
where $\Delta$ is the relative sediment density (-) and $\nu$ is the kinematic viscosity (m$^2$/s).

We added the above information to the methods in our manuscript and also deleted 'applied' (P12, L6-14).

*14/15: How many of these samples came from sections of the flume where no biofilm was present? Without knowing the sampling strategy, this isn't clear. You could use the EPS content values to say something about this.*

None. All of the samples were taken from places in the flume with a visible biofilm present. We clarified our sampling strategy in the main text (P9, L14-21) in response to a comment above. Here, we also explain that the sediment entrainment measurements were destructive and sediment samples for determination of the EPS content could therefore not be taken from the same location. As a consequence, we do not feel that the EPS content values can be used reliably to provide answers on why 42% of the sediment entrainment measurements are smaller than the entrainment threshold of the uncolonised sand.

*15/3: There is clearly more in these data than you currently present, but I agree that these data are not the key point of the paper. However, as these data are compared to the extracted EPS data, it would be helpful to explore them a little more. Could you add the time data onto Fig 2, for example by colouring the points by week? How did variation over time compare to the spatial variation within and between the flumes at any given time? For all the extracted EPS data you plot shear stress against amount of EPS. Why not also plot the natural biofilm data like this?*
There is indeed a wealth of information on natural biofilm behaviour in the data. In another manuscript, we therefore aim to explore the colonisation behaviour of bedforms in more detail because there appears to be an optimum elevation from where the colonisation starts. Also, comparisons with colonisation of other substrates will be made in this other manuscript. As you acknowledge, these detailed colonisation processes are beyond the scope of the current manuscript in which we compare the biostabilisation potential of natural biofilms to that of extracted EPS.

As such, the goal of Figure 2 is to show the probability distribution of the threshold for erosion measurements obtained from the natural biofilm experiment. These data can then be quantitatively compared with the threshold for erosion data obtained from the extracted EPS experiment. The temporal aspect is of smaller importance but, as suggested, we now provided statistics on the weekly measurements. Also, we added some information about the variation in time vs. the variation in space in an effort to explore the spatial and temporal aspects a little more (P17, L4-9).

Regarding your query on plotting shear stress against amount of EPS for the natural biofilm, we now clarified our sampling strategy (see also earlier comments). Since the shear stress and EPS content measurements were not made on the same location, these data cannot be related and plotted in the same manner as was done for the extracted EPS experiments.

*16/6: Make it clear that this section is moving onto the extracted EPS experiments. For example, refer to the 'second set' of experiments (or whatever you call them), rather than just small scale experiments (a phrase that I don't think you've used before).*
Done

*17/1: This paragraph mainly seems to repeat what you have already said in the methods. 17/5 onwards seems to be new information, which should be moved to the methods.*
We removed the repetitive statements and moved the new information to the methods.

*18/Fig. 4 and other figures: Add space between = and following number.*
Done

*18/Table 1: How many replicates?*
5 repeat measurements. We added this information to the manuscript.

*20/Fig 5: A good way of comparing the natural and extracted EPS shear stress values would be to add dashed lines for the mean (and median/standard deviation?) of the shear stresses measured for the natural biofilms. This applies to all figures like this (5/7/8/9).*

*Also, is mixing different to stirring?*
The purpose of figures 5, 7, 8 and 9 is to provide an assessment of the sensitivity of the sediment entrainment threshold of extracted EPS-sand mixtures to the preparation procedure (fig. 5), salinity (fig. 7), pH (fig. 8) and temperature (fig. 9). The highest EPS content (10 g/kg) is used for these sensitivity tests to be able to pick up a difference in entrainment threshold for all these conditions as best as possible. These highest EPS concentrations do, however, not make for a fair comparison with the natural biofilms, since we already know from observations in nature that these EPS contents are on the upper end of the spectrum.

A more meaningful comparison of the shear stresses from natural biofilms and extracted EPS is obtained by contrasting the statistics on shear stress from the natural biofilm experiment with the statistics from the extracted EPS for different contents (as shown in fig 4). This comparison of the shear stresses measured for the natural biofilms and extracted EPS is provided in Table 2.

Stirring and mixing is the same in this context. The procedure is detailed in the methods (P14, 2-10) and therefore removed from the caption of figure 5.

*22/1: Rephrase to clarify that you are referring to the extracted EPS experiments.*
Done
*22/Fig 7: The caption refers to tap water, but distilled water was referred to at the end of 2.3.2.*
This should have been distilled water and we now corrected this.

*26/5: Explain how a mean biostabilisation index is calculated.*
We added an explanation to the methods (P15, L2-10)

*26/13: Quote some figures to support this claim that the index is similar.*
Done

*26/14: Change to 'more suited for replicating'*
Done

*27/Table 2: The caption still doesn't really explain the biostabilisation index. This table could be clearer. It's a bit confusing that the top row of headings doesn't apply all the way down the columns. What statistic is quoted for the extracted EPS results (mean/median)?*
We now added a more detailed explanation of the biostabilisation index to the methods (P15, L2-10). Here, we also define what statistic is used for the extracted EPS results (i.e. mean).

We changed the table so the headings apply all the way down. We considered splitting the data into two separate tables with one for the natural biofilm data and the other for the extracted EPS data, which would have made for simpler tables. However, we believe that it is important to present the data from both experiments in the same table so readers can easily compare the biostabilisation indices obtained in both experiments.

*27/4: Give some values.*
Done

*27/7: I think that you're saying that this technique doesn't measure all of the EPS that is in the sample?*
Yes, that is what we meant to say in a rather clumsy way. We rephrased the sentence to clarify that this is indeed the case (P30, L3-7).

*29/Fig 10: See earlier comment about explaining what transmission is.*
We added this information to the method section, see also earlier comment regarding the transmission and application of the CSM device more generally (P11, L10 – P12, L5).

*30/17: Is the change over time primarily from the sediment drying, rather than any sort of degradation of the extracted EPS?*
We cannot definitively distinguish between both options based on our experiments. It is well known that the sugars in EPS degrade over time and, as such, it is likely that the change over time is attributable to degradation of EPS. Whether the sediment drying counteracts or amplifies this degradation effect does not become clear from the performed experiments. Rather, the performed experiments raise the question of how important sediment saturation is for the behaviour of extracted EPS and we therefore also call for further research on this topic.

[revised manuscript text omitted]

The introduction of the extracted EPS Xanthan Gum in flume experiments investigating bedform dynamics has been shown to change bedform morphology and behaviour (Malarkey et al. 2015; Parsons et al. 2016). Changes in delta morphology and behaviour were also observed in flume experiments where extracted EPS was added to the sediment mixture (Hoyal and Sheets 2009; Kleinhans et al.
20 2014). Extracted EPS are here defined as polysaccharides with a variety of uses (e.g. food additives) that can be extracted from simple sugars using a fermentation process. Extracted EPS are

generally available as a powder and are in this study employed to systematically introduce biological cohesion into physical models., evidence is growing that biofilms alter their local environment by affecting hydrodynamics (Vignaga et al. 2013), since the biofilm surface changes the bed roughness to either dampen or increase turbulence production (Gerbersdorf and Wieprecht 2015), and sometimes their protruding structures create a buffer layer between the flow and the sediment bed that can enhance settling rates (e.g. Augspurger and Küsel 2010).

The corollary of the evidence showing the impact of biofilms on sediment stability and flow behaviour is that the inclusion of biological processes and responses is critical to modelling sediment dynamics because micro-organisms are an integral component of the functioning of water and sediment transfer systems... Predicting the potential impacts of climate change on aquatic environments and applying bio-engineering adaptation strategies like '*Building with Nature*' for coastal defence (de Vriend et al. 2015) or flood resilience (Temmerman et al. 2013) requires an understanding of i) the response of micro-organisms to changes in climate-induced hydrodynamic forcing, and ii) the role of micro-organisms in water and sediment transfer systems. Even though it has been demonstrated that the extracted EPS Xanthan Gum is not a perfect analogue for natural biofilms (Perkins et al. 2004), it isTo date, quantification of biostabilisation effects in space and time remain scarce however. A controlled physical model experiment is therefore employed to systematically investigate and provide further quantification of natural biostabilisation effects. Additionally, the extracted EPS Xantham Gum has proven useful for modelling biological interactions with sediment dynamics (e.g. Hoyal and Sheets 2009; Kleinhans et al. 2014; Malarkey et al. 2015; Parsons et al. 2016). Extracted EPS also has, even though it has been

demonstrated that Xanthan Gum is not a perfect analogue for natural biofilms (Perkins et al. 2004). Extracted EPS generally also have the potential advantages over growing natural biofilms that preparation time and experiment duration in physical models can be reduced and biostabilisation effects can be controlled. In assessing the potential of four extracted EPS to mimic natural biostabilisation, the natural biofilm physical experiment is compared to the complementary experiments on extracted EPS.

The objective of this study is therefore to evaluate biostabilisation effects of existing extracted EPS for a range of conditions commonly used in physical modelling experiments. Two sets of experiments are being reported on: the first set of experiments focusses on biostabilisation resulting from colonisation of sandy substrate by natural biofilms ('natural beds'). The second set of experiment focusses on biostabilisation resulting from the systematic addition of extracted EPS to the same sandy substrate ('engineered beds'). In doing so, the study solely focusses on the sediment stabilising aspect of biofilms and does not explicitly intend to replicate and evaluate natural biofilm behaviour and effects. The novel outcome of this study is the development of a robust methodology and protocol for the application and resultant impacts of extracted EPS, which can be applied to future experimental studies that require the representation of biological cohesion in a rapid and controlled manner. A sandy substrate was used in this study sincebecause this grain size range is most commonly used in physical models of coastal and fluvial systems to date. The specific aims of this study are to:

1. Quantify biostabilisation effects (i.e. erosion threshold for erosion) of natural diatom biofilm-secreted EPS on sandy substrates in a physical model experiment. ('natural beds'; first set of experiments).

2. Quantify the biostabilisation effects of four extracted EPS using the same sandy substrate ('engineered beds'; second set of experiments).

[revised manuscript text omitted]

    1. An initial horizontal line where the transmission is close to 100%.

2. A slope representing the drop in transmission of light across the measurement chamber as erosion occurs and sediment is being suspended.

3. An asymptotic part where the transmission approaches 0 when the air pressure increases.

These profiles vary depending on the sediment properties. Following Tolhurst et al. (1999), the critical erosion threshold was defined as the pressure step at which the transmission drops below 90%.

We calculated the theoretical entrainment threshold $\tau_c$ for our sediment according:

$$\tau_c = \theta_c \cdot (\rho_s - \rho) \cdot g \cdot D_{50} \tag{2}$$

where $\theta_c$ is the Shields number (N/m$^2$), $\rho_s$ is density of sediment (kg/m$^3$), $\rho$ is the density of water (kg/m$^3$) and $D_{50}$ is the median grain size (m). The Shields number $\theta_c$ is calculated following Zanke (2003):

$$\theta_c = 0.145 \cdot \mathrm{Re}_p^{-0.33} + 0.045 \cdot 10^{-1100 \cdot \mathrm{Re}_p^{-1.5}} \tag{3}$$

where $Re_p$ is the Reynolds particle number calculated by:

$$\mathrm{Re}_p = D_{50}^{1.5} \cdot \frac{\sqrt{\Delta \cdot g}}{\nu} \tag{4}$$

where $\Delta$ is the relative sediment density (-) and $\nu$ is the kinematic viscosity (m$^2$/s).

**2.3 Petri dish sediment sample tests with extracted EPS**

In the second set of experiments, the effect of varying amounts of four different types of extracted EPS on the sediment entrainment threshold and erosion behaviour was tested. The four different EPS Xanthan Gum, Alginic Acid, Carrageenan and Agar were selected for their ease of availability,

differences in chemical properties, and absence of safety issues ensuring the potential for wide usage in future work. Xanthan Gum ($C_{35}H_{49}O_{29}$) is a polysaccharide commonly used as a food additive and has also been included in earlier laboratory tests (Tolhurst et al. 2002; Parsons et al. 2016). Alginic Acid ($C_6H_8O_6$)$_n$, also known as alginate, is a carbohydrate produced by brown algae and

5 also widely used in food. Carrageenan is a sulphate polysaccharide extracted from red seaweeds and also widely used as a food additive. We used the Iota variety that has two sulphate groups per disaccharide ($C_{24}H_{36}O_{25}S_2$). Agar is used as a gelling agent and is obtained from the polysaccharide agarose found in some species of red algae.

10 A protocol similar to the one used in Tolhurst et al. (2002) was applied to prepare the petri dish sediment samples for CSM testing. A control test with no EPS, and four tests with increasing EPS contents of 1.25 g, 2.5 g, 5 g and 10 g per kg of sediment were performed for the four different EPS. The applied concentrations of the extracted EPS were based on reported values in the literature (Taylor et al. 1999; Tolhurst et al. 2002) and were also compared to the EPS content measured

15 in the natural biofilm experiment. The required EPS amount was added to 330 ml of distilled water and mixed thoroughly by a magnetic stirrer. The EPS solution was then added to 650 g of dry 110 micron sand and mixed with an electric stirrer to distribute the EPS solution throughout the sand. The sand-EPS mixture was then poured into plastic petri dishes (5 cm diameter) to a depth of 1 cm. Irregularities on the sediment surface increase the bed roughness and stress (Tolhurst et al. 2002),

20 therefore care was taken to create a level surface by tapping the side of the petri dishes before testing.

All test conditions were repeated five times and all tests were performed under fully saturated conditions.

**2.3.1 Preparation procedure**

Protocol development on the application and effects of different extracted EPS required an assessment

5 of the impact of the preparation procedure on the sediment entrainment threshold. To this end, the preparation procedure described above, referred to as 'Wet Mixing', was complemented by a preparation procedure referred to as Dry Mixing. Both procedures used the same sand, EPS and amounts but the order in which they were combined and mixed, was changed. In contrast to the Wet Mixing procedure, in the Dry Mixing procedure the required amount of EPS was first added to the sand

10 and mixed with an electric stirrer. Then, 330 ml of distilled water was added to the dry sand-EPS mixture and a further mixing with the electrical stirrer was performed. Note that the risk of dust formation and associated loss of EPS powder was greater in the Dry Mixing procedure.

**2.3.2 Environmental conditions**

Protocol development on the application and effects of different EPS also required an assessment of the

15 impact of the different environmental conditions on the sediment entrainment threshold. Temperature, salinity and  pH commonly vary between flume facilities. Therefore, a sensitivity analysis on the effects of these environmental conditions on the sediment entrainment threshold for the four extracted EPS was performed. For temperature, tests were performed at 10° C and 40 ° C in addition to the

control tests at room temperature of 20° CelsiusC. For pH, tests were performed with a pH of 4 and a pH of 10 in addition to the control tests of a pH of 7. Standard and commercially available buffer solutions were used to obtain liquids with these pHs. For salinity, tests with a salinity of 30 ppm corresponding to brackish conditions were performed in addition to the control tests with distilled fresh water.

**2.3 Biostabilisation index**

A biostabilisation index (Manzenrieder, 1983; Tolhurst et al., 1999; Friend et al., 2003a; Thom et al., 2010) was calculated to quantify and compare the degree of biostabilisation in the natural biofilm and extracted EPS experiments. The biostabilisation index was calculated from the ratio of the critical erosion shear stress ($\tau_c$) of the relevant experiment, to the $\tau_c$ for the uncolonised sand. Since the same sand was used in both experiments, a direct comparison between biostabilisation indices from the natural biofilm experiment and the extracted EPS can be made. For the natural biofilm experiment, the mean, median and maximum critical erosion shear stresses from 61 measurements were used in calculating the biostabilisation index. For the extracted EPS experiment, the mean critical erosion shear stress was used in calculating the biostabilisation index.

**3 Results**

**3.1 Biofilm colonisation and species ecology of the natural biofilm experiment**

[revised manuscript text omitted]

**Average ± St. deviation erosion threshold for erosion (N·m⁻²)**

| EPS (g·kg⁻¹) | Xanthan Gum | Carrageenan | Agar | Alginic Acid |
|---|---|---|---|---|
| **0** | 0.18 ± 0.06 | 0.18 ± 0.06 | 0.18 ± 0.06 | 0.18 ± 0.06 |

| | | | | |
|---|---|---|---|---|
| **1.25** | 0.32 ± 0.11 | 0.11 ± 0.08 | 0.07 ± 0.06 | 0.11 ± 0.08 |
| **2.5** | 0.87 ± 0.29 | 0.27 ± 0.06 | 0.04 ± 0.03 | 0.09 ± 0.11 |
| **5** | 1.57 ± 0.13 | 0.63 ± 0.04 | 0.03 ± 0.02 | 0.07 ± 0.08 |
| **10** | 3.01 ± 0.21 | 1.36 ± 0.24 | 0.02 ± 0.01 | 0.11 ± 0.10 |

**3.23.2 Effects of preparation procedure on sediment stability**

The preparation procedure adopted for adding the extracted compounds to the sediment material had an impact on the resultant erosion threshold for erosion (Figure 5). 'Dry mixing' the extracted EPS powder with the sediment prior to adding water resulted in a higher erosion threshold for erosion than 'Wet mixing' the EPS powder with sediment in water for all tested EPS. The difference was greatest for Xanthan Gum with a 67% higher threshold for erosion for the dry mixing procedure compared to the wet mixing procedure.

[Figure]

**Figure 5. The erosion thresholds as a function of the preparation procedure for four surrogates as measured with the CSM erosion device.**  **Error bars are standard deviation from n = 5 repeat measurements.**

**3.3.3 Temporal effects on sediment stability**

Time elapsed from initial mixing also affected the sediment stabilising capacity of extracted EPS (Figure 6). Repeat measurements after one day, seven days and fifteen days demonstrated that the  thresholds for erosion remained constant throughout the first week. However, the repeat measurements after fifteen days showed a decrease in the  threshold for erosion below the

 threshold _for erosion_ of sand without EPS. This effectively meant that after about two weeks of

initial application of EPS the impact on the  threshold _for erosion_ of the sediment ceased to exist.

Met opmaak: Engels (V.S.)

[Figure]

**Figure 6. The threshold for  as a function of time for Xanthan Gum and Carrageenan as measured with the**

**CSM erosion device. Error bars are standard deviation from n = 3 repeat measurements.**

Met opmaak: Engels (V.S.)

**3.3.4 Effects of salinity on sediment stability**

Salinity had a limited effect on the threshold for  (Figure 7). Saline water tended to

decrease the threshold for  compared to freshwater conditions, though the differences

are statistically insignificant for all four EPS. The threshold for of Alginic Acid

and Agar remained below the threshold for  of sand without EPS independent of the

Met opmaak: Engels (V.S.)

Met opmaak: Engels (V.S.)

Met opmaak: Engels (V.S.)

Met opmaak: Engels (V.S.)

salinity of the water. These findings imply that the results of the extracted EPS experiments, which were mostly obtained for freshwater conditions, can be extrapolated to saline conditions.

[Figure]

Figure 7. The threshold for erosion  as a function of salinity for four extracted EPS as measured with the CSM erosion device. Distilled water was used for the freshwater tests and a salinity of 30 ppt was used for the saline water tests. The horizontal lines correspond to the threshold for erosion  of sand without EPS for freshwater (dashed) and saline water (dotted). Error bars are standard deviation from n = 3 repeat measurements.

10  ### 3.3.5 Effects of pH on sediment stability

The pH of the applied solution had variable effects on the  threshold for erosion (Figure 8). An acid solution with a pH of 4 resulted in a higher  threshold for erosion of Xanthan Gum, but in a

lower threshold for Carrageenan. An alkaline solution with a pH of 10 resulted in a lower threshold for erosion  of Xanthan Gum as well as Carrageenan. The threshold for erosion  of Alginic Acid and Agar remained below the threshold for erosion  of sand without EPS, independent of the pH of the solution.

[Figure]

**Figure 8. The threshold for erosion  as a function of pH for four extracted EPS as measured with the CSM erosion device. The horizontal lines correspond to the threshold for erosion  of sand without EPS for water with a pH of 7 (dashed), a pH of 4 (dotted), and a pH of 10 (dash-dotted). Error bars are standard deviation from n = 3 repeat measurements.**

**3.3.6 Effects of temperature on sediment stability**

A lower temperature of 10° C as well as a higher temperature of 40° C resulted in a lower  threshold for erosion (Figure 9). For Xanthan Gum as well as Carrageenan, the threshold for erosion was about halved during 10° C and 40° C test conditions compared with 20° C test conditions. The threshold for erosion of Alginic Acid and Agar remained below the  threshold for erosion of sand without EPS independent of the temperature.

[Figure]

**Figure 9. The threshold for erosion thresholds as a function of temperature for four extracted EPS as measured with the CSM erosion device. The horizontal lines correspond to the threshold for erosion thresholds of sand without EPS for a temperature of 20° CelsiusC (dashed), a temperature of 10° CelsiusC (dotted), and a temperature of 40° CelsiusC (dash-dotted). Error bars are standard deviation from n = 3 repeat measurements.**

**3.23.7 Synthesis of the effects of extracted EPS on sediment stability**

In summary, extracted EPS Xanthan Gum and Carrageenan increased the erosion threshold for erosion with higher EPS content (Table 1). For these two EPS, the relation between erosion threshold for erosion and EPS content was linearly and predictable (Figure 4). In contrast, the extracted EPS Alginic Acid and Agar did not increase the erosion threshold for erosion (Table 1), independent of the applied concentration (Figure 4), preparation procedure (Figure 5) or environmental condition such as salinity, pH and temperature. Yet, this study demonstrated that the preparation procedure, environmental

conditions and time impacted on the resultant  threshold for erosion of the EPS Xanthan Gum and Carrageenan. A dry mixing procedure increased the  threshold for erosion while saline water, alkaline solutions and non-room temperature test conditions of 10° C and 40° C decreased the threshold for erosion . The tests also showed that the effects of adding Xanthan

5 Gum and Carrageenan on the threshold for erosion  ceased to exist after about two weeks following initial application (Figure 6). These findings indicate that the effectiveness of extracted EPS to stabilise sediment is sensitive to the applied concentration, the preparation procedure, time and environmental conditions.

**4 Discussion**

10 The CSM data show that addition of extracted EPS Xanthan Gum and Carrageenan increases the critical  threshold for erosion, even at low EPS concentrations (Figure 4 and Table 1). The observation that the  threshold for erosion increased approximately linear with EPS content for Xanthan Gum is in agreement with the findings reported in Tolhurst et al. (2002). We find a similar linear increase in  threshold for erosion with EPS content for Carrageenan, though the

15 rate of increase is smaller compared to Xanthan Gum. The approximately linear relation between EPS content and  threshold for erosion across the measured range for Xanthan Gum and Carrageenan simplifies the prediction of biostabilisation effects due to extracted EPS. Two other extracted EPS, Alginic Acid and Agar, were also tested and showed negligible biostabilisation for any of the test conditions investigated.

Biostabilisation of the same sandy substrate due to natural biofilm colonisation and due to addition of extracted EPS Xanthan Gum and Carrageenan compares well (Table 2). We find a mean biostabilisation index due to natural biofilm colonisation  of almost four times that of the uncolonised sand. Such a biostabilisation index is within the reported range for fine sand (Dade et al. 1990; Vignaga

5  et al. 2013). More specifically, 42% of the tested samples did not show biostabilisation compared to uncolonised sand while 10% of the measurements showed a tenfold biostabilisation relative to uncolonised sand (Figure 2). The presented cumulative probability distribution of critical threshold for erosion  reflects the large spatial and temporal variations generally seen in natural biostabilised environments (Paterson 1989; Amos et al. 1998; Tolhurst et al. 1999,

10  2003; Friend et al. 2003a. For the second set of experiments focusing on extracted EPS, we find similar biostabilisation indices as observed in the first set of experiments on natural biofilms (Table 2). For Xanthan Gum, the biostabilisation index of

15  1.7 for the lowest concentration of 1.25 g·kg$^{-1}$ compares well to the median biostabilisation index of 1.3 in the natural biofilm experiment. The biostabilisation index of 16.4 for the highest concentration of 10 g·kg$^{-1}$ represents the 97[th] percentile of the biostabilisation index of the natural biofilm experiment, and is close to the maximum biostabilisation index of 21. For Carrageen, the biostabilisation indices are

20  generally lower and the biostabilisation index of 1.5 for the concentration of 2.5 g·kg$^{-1}$ compares well to the median biostabilisation index of 1.3 in the natural biofilm experiment. The biostabilisation index of

3.5 for the concentration of 5 g·kg$^{-1}$ is close to the mean biostabilisation index of 3.8 in the natural biofilm experiment. Xanthan Gum may be more suited for replicating the higher biostabilisation observations of natural biofilms due to the higher threshold for erosion of the highest applied contentration of 10 g·kg$^{-1}$. Application of carrageenan may be more appropriate to replicate the lower biostabilisation observations of natural biofilms due to the small effect on the threshold for erosion thresholds for of low concentrations.

**Table 2**.

. Biostabilisation index resulting from natural biofilm colonisation and the addition of Xanthan Gum and Carrageenan extracted EPS to sand.

| |  | *Natural biofilm experiment*  | *Extracted EPS experiment*  |  |
|---|---|---|---|---|
|  |  |  |  |  |

| | Bare sand | Mean | Median | Max. | 1.25 g·kg$^{-1}$ | 2.5 g·kg$^{-1}$ | 5 g·kg$^{-1}$ | 10 g·kg$^{-1}$ |
|---|---|---|---|---|---|---|---|---|
|  | |  |  |  | | |  | |
|  | |  |  |  | | |  | |

|  | | | | | *Wet mix* | *Dry mix* | *Saline* | *pH = 10* | *T = 10° Celsius*  |
|---|---|---|---|---|---|---|---|---|---|
| **Natural biofilm** | 1 | 1.3 | 3.8 | 21.0 | - | - | - | - | - | - | - | - |

| | | | | | | | | | | |
|---|---|---|---|---|---|---|---|---|---|---|
| **Xanthan Gum** | 1 | - | - | - | 1.7 | 4.8 | 8.6 | 16.4 | 27.6 | 15.2 | 10.3 | 7.8 |
| **Carrageenan** | 1 | - | - | - | 0.6 | 1.5 | 3.5 | 7.4 | 9.8 | 4.7 | 2.2 | 1.6 |

The concentrations of the EPS derived from the natural biofilm experiment (Figure 3, ~8 µg g$^{-1}$) are about three orders of magnitude lower than the applied extracted EPS concentrations (2.5 – 10 mg g$^{-1}$) to achieve the same biostabilisation effect (Table 2). Two reasons may explain these differences. First, the applied phenol-sulphuric acid assay only measures a carbohydrate fraction of the total EPS as well as some low-weight sugars that are extracted with the polymeric material (Underwood et al. 1995). As a result, this technique may not measure all of the EPS present in the sample, and is also known to be sensitive to a host of conditions (Perkins et al. 2004). Second, sediment sampling for EPS concentration analysis typically involved scraping off the top centimetre of the substrate. However, it has been shown that EPS content in nature is highest at the sediment surface (top 200 µm) and decreases with depth (Taylor and Paterson 1998). Our sediment sampling strategy is likely to have diluted the EPS concentration, which may offer another explanation for the lower EPS concentrations in the natural biofilm samples.

[Figure]

**Figure 10. CSM erosion profiles for sediment with different degrees of biostability due to natural biofilm colonisation (A) and due to different Xanthan Gum and Carrageenan extracted EPS contents (B). Following Tolhurst et al. (1999), the eroding pressure corresponding to a 90% transmission is defined as the erosion event.**

[revised manuscript text omitted]